# Provable benefits of score matching

**Chirag Pabbaraju**
**Stanford University**
cpabbara@cs.stanford.edu

**Dhruv Rohatgi**
**MIT**
drohatgi@mit.edu

**Anish Sevekari**
**Carnegie Mellon University**
asevekar@andrew.cmu.edu

**Holden Lee**
**Johns Hopkins University**
hlee283@jhu.edu

**Ankur Moitra**
**MIT**
moitra@mit.edu

**Andrej Risteski**
**Carnegie Mellon University**
aristesk@andrew.cmu.edu

## Abstract

Score matching is an alternative to maximum likelihood (ML) for estimating a probability distribution parametrized up to a constant of proportionality. By fitting the "score" of the distribution, it sidesteps the need to compute this constant of proportionality (which is often intractable). While score matching and variants thereof are popular in practice, precise theoretical understanding of the benefits and tradeoffs with maximum likelihood—both computational and statistical—are not well understood. In this work, we give the first example of a natural exponential family of distributions such that the score matching loss is computationally efficient to optimize, and has a comparable statistical efficiency to ML, while the ML loss is intractable to optimize using a gradient-based method. The family consists of exponentials of polynomials of fixed degree, and our result can be viewed as a continuous analogue of recent developments in the discrete setting. Precisely, we show: (1) Designing a zeroth-order or first-order oracle for optimizing the maximum likelihood loss is NP-hard. (2) Maximum likelihood has a statistical efficiency polynomial in the ambient dimension and the radius of the parameters of the family. (3) Minimizing the score matching loss is both computationally and statistically efficient, with complexity polynomial in the ambient dimension.

## 1 Introduction

Energy-based models are a flexible class of probabilistic models with wide-ranging applications. They are parameterized by a class of energies $E_\theta(x)$ which in turn determines the distribution

$$p_\theta(x) = \frac{\exp(-E_\theta(x))}{Z_\theta}$$

up to a constant of proportionality $Z_\theta$ that is called the partition function. One of the major challenges of working with energy-based models is designing efficient algorithms for fitting them to data. Statistical theory tells us that the maximum likelihood estimator (MLE)—i.e., the parameters $\theta$ which maximize the likelihood—enjoys good statistical properties including consistency and asymptotic efficiency.

However, there is a major computational impediment to computing the MLE: Both evaluating the log-likelihood and computing its gradient with respect to $\theta$ (i.e., implementing zeroth and first order oracles, respectively) seem to require computing the partition function, which is often computationally intractable. More precisely, the gradient of the negative log-likelihood depends on $\nabla_\theta \log Z_\theta = \mathbb{E}_{p_\theta}[\nabla_\theta E_\theta(x)]$. A popular approach is to estimate this quantity by using a Markov chain to approximately sample from $p_\theta$. However in high-dimensional settings, Markov chains often require many, sometimes even exponentially many, steps to mix.

Score matching (Hyvärinen, 2005) is a popular alternative that sidesteps needing to compute the partition function of sample from $p_\theta$. The idea is to fit the score of the distribution, in the sense that

37th Conference on Neural Information Processing Systems (NeurIPS 2023).

we want $\theta$ such that $\nabla_x \log p(x)$ matches $\nabla_x \log p_\theta(x)$ for a typical sample from $p$. This approach turns out to have many nice properties. It is consistent in the sense that minimizing the objective function yields provably good estimates for the unknown parameters. Moreover, while the definition depends on the unknown $\nabla_x \log p(x)$, by applying integration by parts, it is possible to transform the objective into an equivalent one that can be estimated from samples.

The main question is to bound its statistical performance, especially relative to that of the maximum likelihood estimator. Recent work by Koehler et al. (2022) showed that the cost can be quite steep. They gave explicit examples of distributions that have bad isoperimetric properties (i.e., large Poincaré constant) and showed how such properties can cause poor statistical performance.

Despite wide usage, there is little rigorous understanding of when score matching *helps*. This amounts to finding a general setting where maximizing the likelihood with standard first-order optimization is provably hard, and yet score matching is both computationally and statistically efficient, with only a polynomial loss in sample complexity relative to the MLE. In this work, we show the first such guarantees, and we do so for a natural class of exponential families defined by polynomials. As we discuss in Section 1.1, our results parallel recent developments in learning graphical models—where it is known that pseudolikelihood methods allow efficient learning of distributions that are hard to sample from—and can be viewed as a continuous analogue of such results.

In general, an exponential family on $\mathbb{R}^n$ has the form $p_\theta(x) \propto h(x) \exp(\langle \theta, T(x) \rangle)$ where $h(x)$ is the *base measure*, $\theta$ is the *parameter vector*, and $T(x)$ is the vector of *sufficient statistics*. Exponential families are one of the most classic parametric families of distributions, dating back to works by Darmois (1935), Koopman (1936) and Pitman (1936). They have a number of natural properties, including: (1) The parameters $\theta$ are uniquely determined by the expectation of the sufficient statistics $\mathbb{E}_{p_\theta}[T]$; (2) The distribution $p_\theta$ is the maximum entropy distribution, subject to having given values for $\mathbb{E}_{p_\theta}[T]$; (3) They have conjugate priors (Brown, 1986), which allow characterizations of the family for the posterior of the parameters given data.

For any (odd positive integer) constant $d$ and norm bound $B \geq 1$, we study a natural exponential family $\mathcal{P}_{n,d,B}$ on $\mathbb{R}^n$ where

1. The *sufficient statistics* $T(x) \in \mathbb{R}^{M-1}$ consist of all monomials in $x_1, \ldots, x_n$ of degree at least 1 and at most $d$ $\left(\text{where } M = \binom{n+d}{d}\right)$.
2. The *base measure* is defined as $h(x) = \exp(-\sum_{i=1}^{n} x_i^{d+1})$.[1]
3. The *parameters* $\theta$ lie in an $l_\infty$-ball: $\theta \in \Theta_B = \{\theta \in \mathbb{R}^{M-1} : \|\theta\|_\infty \leq B\}$.

Towards stating our main results, we formally define the maximum likelihood and score matching objectives, denoting by $\hat{\mathbb{E}}$ the empirical average over the training samples drawn from some $p \in \mathcal{P}_{n,d,B}$:

$$L_{\mathbf{MLE}}(\theta) = \hat{\mathbb{E}}_{x \sim p}[\log p_\theta(x)]$$

$$L_{\mathbf{SM}}(\theta) = \frac{1}{2}\hat{\mathbb{E}}_{x \sim p}[\|\nabla \log p(x) - \nabla \log p_\theta(X)\|^2] + K_p$$

$$= \hat{\mathbb{E}}_{x \sim p}\left[\operatorname{Tr} \nabla^2 \log p_\theta(x) + \frac{1}{2}\|\nabla \log p_\theta(x)\|^2\right] \quad (1)$$

where $K_p$ is a constant depending only on $p$ and (1) follows by integration by parts (Hyvärinen, 2005). In the special case of exponential families, (1) is a quadratic, and in fact the optimum can be written in closed form:

$$\arg\min_\theta L_{\mathbf{SM}}(\theta) = -\hat{\mathbb{E}}_{x \sim p}[(JT)_x(JT)_x^T]^{-1}\hat{\mathbb{E}}_{x \sim p}\Delta T(x) \quad (2)$$

where $(JT)_x : (M-1) \times n$ is the Jacobian of $T$ at the point $x$, $\Delta f = \sum_i \partial_i^2 f$ is the Laplacian, applied coordinate wise to the vector-valued function $f$.

With this setting in place, we show the following intractability result.

**Theorem 1.1** (Informal, computational lower bound). *Unless RP = NP, there is no* $\operatorname{poly}(n, N)$-*time algorithm that evaluates* $L_{\mathbf{MLE}}(\theta)$ *and* $\nabla L_{\mathbf{MLE}}(\theta)$ *given* $\theta \in \Theta_B$ *and arbitrary samples* $x_1, \ldots, x_N \in \mathbb{R}^n$, *for* $d = 7, B = \operatorname{poly}(n)$. *Thus, optimizing the MLE loss using a zeroth-order or first-order method is computationally intractable.*

---

[1]We note that the choice of base measure is for convenience in ensuring tail bounds necessary in our proof.

The main idea of the proof is to construct a polynomial $F_{\mathcal{C}}(x)$ which has roots exactly at the satisfying assignments of a given 3-SAT formula $\mathcal{C}$. We then argue that $\exp(-\gamma F_{\mathcal{C}}(x))$, for sufficiently large $\gamma > 0$, concentrates near the satisfying assignments. Finally, we show sampling from this distribution or approximating $\log Z_\theta$ or $\nabla_\theta \log Z_\theta$ (where $\theta \in \mathbb{R}^{M-1}$ is the parameter vector corresponding to the polynomial $-\gamma F_{\mathcal{C}}(x)$) would enable efficiently finding a satisfying assignment.

Our next result shows that MLE, though computationally intractable to compute via implementing zeroth or first order oracles, has (asymptotic) sample complexity $\mathrm{poly}(n, B)$ (for constant $d$).

**Theorem 1.2** (Informal, efficiency of MLE). *The MLE estimator $\hat{\theta}_{\mathrm{MLE}} = \arg\max_\theta L_{\mathrm{MLE}}(\theta)$ has asymptotic sample complexity polynomial in $n$. That is, for all sufficiently large $N$ it holds with probability at least $0.99$ (over $N$ samples drawn from $p_{\theta*}$) that:*

$$\|\hat{\theta}_{\mathrm{MLE}} - \theta^*\|^2 \leq O\left(\frac{(nB)^{\mathrm{poly}(d)}}{N}\right).$$

The main proof technique for this is an anticoncentration bound of low-degree polynomials, for distributions in our exponential family.

Lastly, we prove that score matching *also* has polynomial (asymptotic) statistical complexity.

**Theorem 1.3** (Informal, efficiency of SM). *The score matching estimator $\hat{\theta}_{\mathrm{SM}} = \arg\max_\theta L_{\mathrm{SM}}(\theta)$ also has asymptotic sample complexity at most polynomial in $n$. That is, for all sufficiently large $N$ it holds with probability at least $0.99$ (over $N$ samples drawn from $p_{\theta*}$) that:*

$$\|\hat{\theta}_{\mathrm{SM}} - \theta^*\|^2 \leq O\left(\frac{(nB)^{\mathrm{poly}(d)}}{N}\right). \tag{3}$$

The main ingredient in this result is a bound on the *restricted Poincaré constant*—namely, the Poincaré constant, when restricted to functions that are linear in the sufficient statistics $T$. We bound this quantity for the exponential family we consider in terms of the condition number of the Fisher matrix of the distribution, which we believe is a result of independent interest. With this tool in hand, we can use the framework of Koehler et al. (2022), which relates the asymptotic sample complexity of score matching to the asymptotic sample complexity of maximum likelihood, in terms of the restricted Poincaré constant of the distribution.

## 1.1 Discussion and related work

**Score matching:** Score matching was proposed by Hyvärinen (2005), who also gave conditions under which it is consistent and asymptotically normal. Asymptotic normality is also proven for various kernelized variants of score matching in Barp et al. (2019). Koehler et al. (2022) prove that the statistical sample complexity of score matching is not much worse than the sample complexity of maximum likelihood when the distribution satisfies a (restricted) Poincaré inequality. While we leverage machinery from Koehler et al. (2022), their work only bounds the sample complexity of score matching by a quantity polynomial in the ambient dimension for a specific distribution in a specific bimodal exponential family. By contrast, we can handle an entire class of exponential families with low-degree sufficient statistics.

**Poincaré vs Restricted Poincaré:** We note that while Poincaré inequalities are directly related to isoperimetry and mixing of Markov chains, sample efficiency of score matching only depends on the Poincaré inequality holding for a *restricted* class of functions, namely, functions linear in the sufficient statistics. Hence, hardness of sampling only implies sample complexity lower bounds in cases where the family is expressive enough—indeed, the key to exponential lower bounds for score matching in Koehler et al. (2022) is augmenting the sufficient statistics with a function defined by a bad cut. This gap means that we can hope to have good sample complexity for score matching even in cases where sampling is hard—which we take advantage of in this work.

**Learning exponential families:** Despite the fact that exponential families are both classical and ubiquitous, both in statistics and machine learning, there is relatively little understanding about the computational-statistical tradeoffs to learn them from data, that is, what sample complexity can be achieved with a computationally efficient algorithm. Ren et al. (2021) consider a version of the

"interaction screening" estimator, a close relative of pseudolikelihood, but do not prove anything about the statistical complexity of this estimator. Shah et al. (2021) consider a related estimator, and analyze it under various low-rank and sparsity assumptions of reshapings of the sufficient statistics into a tensor. Unfortunately, these assumptions are somewhat involved, and it's unclear if they are needed for designing computationally and statistically efficient algorithms.

**Discrete exponential families (Ising models):** Ising models have the form $p_J(x) \propto \exp\left(\sum_{i \sim j} J_{ij} x_i x_j + \sum_i J_i x_i\right)$ where $\sim$ denotes adjacency in some (unknown) graph, and $J_{ij}, J_i$ denote the corresponding pairwise and singleton potentials. Bresler (2015) gave an efficient algorithm for learning any Ising model over a graph with constant degree (and $l_\infty$-bounds on the coefficients); see also the more recent work (Dagan et al., 2021). In contrast, it is a classic result (Arora and Barak, 2009) that approximating the partition function of members in this family is NP-hard.

Similarly, the exponential family we consider is such that it contains members for which sampling and approximating their partition function is intractable (the main ingredient in the proof of Theorem 1.1). Nevertheless, by Theorem 3, we can learn the parameters for members in this family computationally efficiently, and with sample complexity comparable to the optimal one (achieved by maximum likelihood). This also parallels other developments in Ising models (Bresler et al., 2014; Montanari, 2015), where it is known that restricting the type of learning algorithm (e.g., requiring it to work with sufficient statistics only) can make a tractable problem become intractable.

The parallels can be drawn even on an algorithmic level: a follow up work to Bresler (2015) by Vuffray et al. (2016) showed that similar results can be shown in the Ising model setting by using the "screening estimator", a close relative of the classical pseudolikelihood estimator (Besag, 1977) which tries to learn a distribution by matching the conditional probability of singletons, and thereby avoids having to evaluate a partition function. Since conditional probabilities of singletons capture changes in a single coordinate, they can be viewed as a kind of "discrete gradient"—a further analogy to score matching in the continuous setting.[2]

## 2 Preliminaries

We consider the following exponential family. Fix positive integers $n, d, B \in \mathbb{N}$ where $d$ is odd. Let $h(x) = \exp(-\sum_{i=1}^n x_i^{d+1})$, and let $T(x) \in \mathbb{R}^{M-1}$ be the vector of monomials in $x_1, \ldots, x_n$ of degree at least 1 and at most $d$ (so that $M = \binom{n+d}{d}$). Define $\Theta \subseteq \mathbb{R}^{M-1}$ by $\Theta = \{\theta \in \mathbb{R}^{M-1} : \|\theta\|_\infty \leq B\}$. For any $\theta \in \Theta$ define $p_\theta : \mathbb{R}^n \to [0, \infty)$ by

$$p_\theta(x) := \frac{h(x) \exp(\langle \theta, T(x) \rangle)}{Z_\theta}$$

where $Z_\theta = \int_{\mathbb{R}^n} h(x) \exp(\langle \theta, T(x) \rangle) \, dx$ is the normalizing constant. Then we consider the family $\mathcal{P}_{n,d,B} := (p_\theta)_{\theta \in \Theta_B}$. Throughout, we will assume that $B \geq 1$.

**Polynomial notation:** Let $\mathbb{R}[x_1, \ldots, x_n]_{\leq d}$ denote the space of polynomials in $x_1, \ldots, x_n$ of degree at most $d$. We can write any such polynomial $f$ as $f(x) = \sum_{|\mathbf{d}| \leq d} a_\mathbf{d} x_\mathbf{d}$ where $\mathbf{d}$ denotes a degree function $\mathbf{d} : [n] \to \mathbb{N}$, and $|\mathbf{d}| = \sum_{i=1}^n \mathbf{d}(i)$, and we write $x_\mathbf{d}$ to denote $\prod_{i=1}^n x_i^{\mathbf{d}(i)}$. Note that every $\mathbf{d}$ with $1 \leq |\mathbf{d}| \leq d$ corresponds to an index of $T$, i.e. $T(x)_\mathbf{d} = x_\mathbf{d}$.

Let $\|\cdot\|_{\text{mon}}$ denote the $\ell^2$ norm of a polynomial in the monomial basis; that is, $\|\sum_\mathbf{d} a_\mathbf{d} x_\mathbf{d}\|_{\text{mon}} = \left(\sum_\mathbf{d} a_\mathbf{d}^2\right)^{1/2}$. For any function $f : \mathbb{R}^n \to \mathbb{R}$, let $\|f\|_{L^2([-1,1]^n)}^2 = \mathbb{E}_{x \sim \text{Unif}([-1,1]^n)} f(x)^2$.

**Statistical efficiency of MLE:** For any $\theta \in \mathbb{R}^{M-1}$, the Fisher information matrix of $p_\theta$ with respect to the sufficient statistics $T(x)$ is defined as

$$\mathcal{I}(\theta) := \mathbb{E}_{x \sim p_\theta}[T(x) T(x)^\top] - \mathbb{E}_{x \sim p_\theta}[T(x)] \mathbb{E}_{x \sim p_\theta}[T(x)]^\top.$$

It is well-known that for any exponential family with no affine dependencies among the sufficient statistics (see e.g., Theorem 4.6 in Van der Vaart (2000)), it holds that for any $\theta^* \in \mathbb{R}^{M-1}$, given $N$

---

[2]In fact, ratio matching, proposed in Hyvärinen (2007) as a discrete analogue of score matching, relies on exactly this intuition.

independent samples $x^{(1)}, \ldots, x^{(N)} \sim p_{\theta^*}$, the estimator $\hat{\theta}_{\mathrm{MLE}} = \hat{\theta}_{\mathrm{MLE}}(x^{(1)}, \ldots, x^{(N)})$ satisfies

$$\sqrt{N}\left(\hat{\theta}_{\mathrm{MLE}} - \theta^*\right) \to \mathcal{N}(0, \mathcal{I}(\theta^*)^{-1}).$$

**Statistical efficiency of score matching:**   Our analysis of the statistical efficiency of score matching is based on a result due to Koehler et al. (2022). We state a requisite definition followed by the result.

**Definition 2.1** (Restricted Poincaré for exponential families)**.** The restricted Poincaré constant of $p \in \mathcal{P}_{n,d,B}$ is the smallest $C_P > 0$ such that for all $w \in \mathbb{R}^{M-1}$, it holds that

$$\mathrm{Var}_p(\langle w, T(x) \rangle) \leq C_P \mathbb{E}_{x \sim p} \|\nabla_x \langle w, T(x) \rangle\|_2^2.$$

**Theorem 2.2** (Koehler et al. (2022))**.** *Under certain regularity conditions (see Lemma B.4), for any $p_{\theta^*}$ with restricted Poincaré constant $C_P$ and with $\lambda_{min}(\mathcal{I}(\theta^*)) > 0$, given $N$ independent samples $x^{(1)}, \ldots, x^{(N)} \sim p_{\theta^*}$, the estimator $\hat{\theta}_{\mathrm{SM}} = \hat{\theta}_{\mathrm{SM}}(x^{(1)}, \ldots, x^{(N)})$ satisfies*

$$\sqrt{N}(\hat{\theta}_{\mathrm{SM}} - \theta^*) \to \mathcal{N}(0, \Gamma)$$

*where $\Gamma$ satisfies*

$$\|\Gamma\|_{op} \leq \frac{2 C_P^2 (\|\theta^*\|_2^2 \, \mathbb{E}_{x \sim p_{\theta^*}} \|(JT)(x)\|_{op}^4 + \mathbb{E}_{x \sim p_{\theta^*}} \|\Delta T(x)\|_2^2)}{\lambda_{min}(\mathcal{I}(\theta^*))^2}$$

*where $(JT)(x)_i = \nabla_x T_i(x)$ and $\Delta T(x) = \mathrm{Tr}\, \nabla_x^2 T(x)$.*

# 3   Hardness of Implementing Optimization Oracles for $\mathcal{P}_{n,7,\mathrm{poly}(n)}$

In this section we prove NP-hardness of implementing approximate zeroth-order and first-order optimization oracles for maximum likelihood in the exponential family $\mathcal{P}_{n,7,Cn^2 \log(n)}$ (for a sufficiently large constant $C$) as defined in Section 2; we also show that approximate sampling from this family is NP-hard. See Theorems 3.4, 3.5, and A.5 respectively. All of the hardness results proceed by reduction from 3-SAT and use the same construction.

The idea is that for any formula $\mathcal{C}$ on $n$ variables, we can construct a non-negative polynomial $F_{\mathcal{C}}$ of degree at most 6 in variables $x_1, \ldots, x_n$, which has roots exactly at the points of the hypercube $\mathcal{H} := \{-1, 1\}^n \subseteq \mathbb{R}^n$ that correspond to satisfying assignments (under the bijection that $x_i = 1$ corresponds to True and $x_i = -1$ corresponds to False). Intuitively, the distribution with density proportional to $\exp(-\gamma F_{\mathcal{C}}(x))$ will, for sufficiently large $\gamma > 0$, concentrate on the satisfying assignments. It is then straightforward to see that sampling from this distribution or efficiently computing either $\log Z_\theta$ or $\nabla_\theta \log Z_\theta$ (where $\theta \in \mathbb{R}^{M-1}$ is the parameter vector corresponding to the polynomial $-\gamma F_{\mathcal{C}}(x)$) would enable efficiently finding a satisfying assignment.

The remainder of this section makes the above intuition precise; important details include (1) incorporating the base measure $h(x) = \exp(-\sum_{i=1}^n x_i^8)$ into the density function, and (2) showing that a polynomially-large temperature $\gamma$ suffices.

**Definition 3.1** (Clause/formula polynomials)**.** Given a 3-clause formula of the form $C = \tilde{x}_i \vee \tilde{x}_j \vee \tilde{x}_k$ where $\tilde{x}_i = x_i$ or $\tilde{x}_i = \neg x_i$, we construct a polynomial $H_C \in \mathbb{R}[x_1, \ldots, x_n]_{\leq 6}$ defined by

$$H_C(x) = f_i(x_i)^2 f_j(x_j)^2 f_k(x_k)^2$$

where

$$f_i(t) = \begin{cases} (t+1) & \text{if } x_i \text{ is negated in } C \\ (t-1) & \text{otherwise} \end{cases}.$$

For example, if $C = x_1 \vee x_2 \vee \neg x_3$, then $H_C = (x_1 - 1)^2(x_2 - 1)^2(x_3 + 1)^2$. Further, given a 3-SAT formula $\mathcal{C} = C_1 \wedge \cdots \wedge C_m$ on $m$ clauses[3], we define the polynomial

$$H_{\mathcal{C}}(x) = H_{C_1}(x) + \cdots + H_{C_m}(x).$$

It can be seen that any $x \in \mathcal{H}$ corresponds to a satisfying assignment for $\mathcal{C}$ if and only if $H_{\mathcal{C}}(x) = 0$. Note that there are possibly points outside $\mathcal{H}$ which satisfy $H_{\mathcal{C}}(x) = 0$. To avoid these solutions, we introduce another polynomial:

---

[3]It suffices to work with $m = O(n)$, see Theorem A.1.

**Definition 3.2** (Hypercube polynomial)**.** We define $G : \mathbb{R}^n \to \mathbb{R}$ by $G(x) = \sum_{i=1}^n (1 - x_i^2)^2$.

Note that $G(x) \geq 0$ for all $x$, and the roots of $G(x)$ are precisely the vertices of $\mathcal{H}$. Therefore for any $\alpha, \beta > 0$, the roots (in $\mathbb{R}^n$) of the polynomial $F_\mathcal{C}(x) = \alpha H_\mathcal{C}(x) + \beta G(x)$ are precisely the vertices of $\mathcal{H}$ that correspond to satisfying assignments for $\mathcal{C}$.

**Definition 3.3.** Let $\mathcal{C}$ be a 3-CNF formula with $n$ variables and $m$ clauses. Let $\alpha, \beta > 0$. Then we define a distribution $P_{\mathcal{C},\alpha,\beta}$ with density function

$$p_{\mathcal{C},\alpha,\beta}(x) := \frac{h(x) \exp(-\alpha H_\mathcal{C}(x) - \beta G(x))}{Z_{\mathcal{C},\alpha,\beta}}$$

where $Z_{\mathcal{C},\alpha,\beta} = \int_{\mathbb{R}^n} h(x) \exp(-\alpha H_\mathcal{C}(x) - \beta G(x)) \, dx$.

This distribution lies in the exponential family $\mathcal{P}_{n,d,B}$, for $d = 7$ and $B = \Omega(\beta + m\alpha)$ (Lemma A.2). Thus, if $\theta(\mathcal{C}, \alpha, \beta)$ is the parameter vector that induces $P_{\mathcal{C},\alpha,\beta}$, then it suffices to show that (a) approximating $\log Z_{\theta(\mathcal{C},\alpha,\beta)}$, (b) approximating $\nabla_\theta \log Z_{\theta(\mathcal{C},\alpha,\beta)}$, and (c) sampling from $P_{\mathcal{C},\alpha,\beta}$ are NP-hard (under randomized reductions). We sketch the proofs below; details are in Appendix A.

**Hardness of approximating** $\log Z_{\mathcal{C},\alpha,\beta}$**:** In order to prove (a), we bound the mass of $P_{\mathcal{C},\alpha,\beta}$ in each orthant of $\mathbb{R}^n$. In particular, we show that for $\alpha = \Omega(n)$ and $\beta = \Omega(m \log m)$, any orthant corresponding to a satisfying assignment has exponentially larger contribution to $Z_{\mathcal{C},\alpha,\beta}$ than any orthant corresponding to an unsatisfying assignment. A consequence is that the partition function $Z_{\mathcal{C},\alpha,\beta}$ is exponentially larger when the formula $\mathcal{C}$ is satisfiable than when it isn't (Lemma A.6).

But then approximating $Z_{\mathcal{C},\alpha,\beta}$ allows distinguishing a satisfiable formula from an unsatisfiable formula, which is NP-hard. This implies the following theorem (proof in Section A.2):

**Theorem 3.4.** *Fix $n \in \mathbb{N}$ and let $B \geq Cn^2$ for a sufficiently large constant $C$. Unless $\mathsf{RP} = \mathsf{NP}$, there is no $\mathrm{poly}(n)$-time algorithm which takes as input an arbitrary $\theta \in \Theta_B$ and outputs an approximation of $\log Z_\theta$ with additive error less than $n \log 1.16$.*

**Hardness of approximating** $\nabla_\theta \log Z_{\theta(\mathcal{C},\alpha,\beta)}$**:** Note that $\nabla_\theta \log Z_\theta = \mathbb{E}_{x \sim p_\theta}[T(x)]$, so in particular approximating the gradient yields an approximation to the mean $\mathbb{E}_{x \sim p_\theta}[x]$. Since $P_{\mathcal{C},\alpha,\beta}$ is concentrated in orthants corresponding to satisfying assignments of $\mathcal{C}$, we would intuitively expect that if $\mathcal{C}$ has exactly one satisfying assignment $v^*$, then $\mathrm{sign}(\mathbb{E}_{p_\theta}[x])$ corresponds to this assignment. Formally, we show that if $\alpha = \Theta(n)$ and $\beta = \Omega(mn \log m)$, then $\mathbb{E}_{x \sim p_{\mathcal{C},\alpha,\beta}}[v_i^* x_i] \geq 1/20$ for all $i \in [n]$ (Lemma A.7).

Since solving a formula with a unique satisfying assignment is still NP-hard, we get the following theorem (proof in Section A.3):

**Theorem 3.5.** *Fix $n \in \mathbb{N}$ and let $B \geq Cn^2 \log(n)$ for a sufficiently large constant $C$. Unless $\mathsf{RP} = \mathsf{NP}$, there is no $\mathrm{poly}(n)$-time algorithm which takes as input an arbitrary $\theta \in \Theta_B$ and outputs an approximation of $\nabla_\theta \log Z_\theta$ with additive error (in an $l_\infty$ sense) less than $1/20$.*

With the above two theorems in hand, we are ready to present the formal version of Theorem 1.1; the proof is immediate from the definition of $L_{\mathrm{MLE}}(\theta)$ (see Section A.5).

**Corollary 3.6.** *Fix $n, N \in \mathbb{N}$ and let $B \geq Cn^2 \log n$ for a sufficiently large constant $C$. Unless $\mathsf{RP} = \mathsf{NP}$, there is no $\mathrm{poly}(n, N)$-time algorithm which takes as input an arbitrary $\theta \in \Theta_B$, and an arbitrary sample $x_1, \ldots, x_N \in \mathbb{R}^n$, and outputs an approximation of $L_{\mathrm{MLE}}(\theta)$ up to additive error of $n \log 1.16$, or $\nabla_\theta L_{\mathrm{MLE}}(\theta)$ up to an additive error of $1/20$.*

**Hardness of approximate sampling:** We show that for $\alpha = \Omega(n)$ and $\beta = \Omega(m \log m)$, the likelihood that $x \sim P_{\mathcal{C},\alpha,\beta}$ lies in an orthant corresponding to a satisfying assignment for $\mathcal{C}$ is at least $1/2$ (Lemma A.4). Hardness of approximate sampling follows immediately (Theorem A.5). Hence, although we show that score matching can efficiently estimate $\theta^*$ from samples produced by nature, knowing $\theta^*$ isn't enough to efficiently *generate* samples from the distribution.

## 4 Statistical Efficiency of Maximum Likelihood

In this section we prove Theorem 1.2 by showing that for any $\theta \in \Theta_B$, we can lower bound the smallest eigenvalue of the Fisher information matrix $\mathcal{I}(\theta)$. Concretely, we show:

**Theorem 4.1.** *For any $\theta \in \Theta_B$, it holds that*

$$\lambda_{\min}(\mathcal{I}(\theta)) \geq (nB)^{-O(d^3)}.$$

*As a corollary, given $N$ samples from $p_\theta$, it holds as $N \to \infty$ that $\sqrt{N}(\hat{\theta}_{\text{MLE}} - \theta) \to N(0, \Gamma_{\text{MLE}})$ where $\|\Gamma_{\text{MLE}}\|_{op} \leq (nB)^{O(d^3)}$. Moreover, for sufficiently large $N$, with probability at least $0.99$ it holds that $\left\|\hat{\theta}_{\text{MLE}} - \theta\right\|_2^2 \leq (nB)^{O(d^3)}/N$.*

Once we have the bound on $\lambda_{\min}(\mathcal{I}(\theta))$, the first corollary follows from standard bounds for MLE (Section 2), and the second corollary follows from Markov's inequality (see e.g., Remark 4 in Koehler et al. (2022)). Lower-bounding $\lambda_{\min}(\mathcal{I}(\theta))$ itself requires lower-bounding the variance of any polynomial (with respect to $p_\theta$) in terms of its coefficients. The proof consists of three parts. First, we show that the norm of a polynomial in the monomial basis is upper-bounded in terms of its $L^2$ norm on $[-1,1]^n$:

**Lemma 4.2.** *For $f \in \mathbb{R}[x_1, \ldots, x_n]_{\leq d}$, we have $\|f\|_{mon}^2 \leq \binom{n+d}{d}(16e)^d \|f\|_{L^2([-1,1]^n)}^2$.*

The key idea behind this proof is to work with the basis of (tensorized) Legendre polynomials, which is orthonormal with respect to the $L^2$ norm. Once we write the polynomial with respect to this basis, the $L^2$ norm equals the Euclidean norm of the coefficients. Given this observation, all that remains is to bound the coefficients after the change-of-basis. The complete proof is deferred to Appendix C.

Next, we show that if a polynomial $f : \mathbb{R}^n \to \mathbb{R}$ has small variance with respect to $p$, then there is some box on which $f$ has small variance with respect to the uniform distribution. This provides a way of comparing the variance of $f$ with its $L^2$ norm (after an appropriate rescaling).

**Lemma 4.3.** *Fix any $\theta \in \Theta_B$ and define $p := p_\theta$. Define $R := 2^{d+8}nBM$. Then for any $f \in \mathbb{R}[x_1, \ldots, x_n]_{\leq d}$, there is some $z \in \mathbb{R}^n$ with $\|z\|_\infty \leq R$ and some $\epsilon \geq 1/(2(d+1)MR^d(n+B))$ such that*

$$\text{Var}_p(f) \geq \frac{1}{2e} \text{Var}_{\tilde{\mathcal{U}}}(f),$$

*where $\tilde{\mathcal{U}}$ is the uniform distribution on $\{x \in \mathbb{R}^n : \|x - z\|_\infty \leq \epsilon\}$.*

In order to prove this result, we pick a random box of radius $\epsilon$ (within a large bounding box of radius $R$). In expectation, the variance on this box (with respect to $p$) is not much less than $\text{Var}_p(f)$. Moreover, for sufficiently small $\epsilon$, the density function of $p$ on this box has bounded fluctuations, allowing comparison of $\text{Var}_p(f)$ and $\text{Var}_{\tilde{\mathcal{U}}}(f)$. This argument is formalized in Appendix C.

Together, Lemma 4.2 and 4.3 allow us to lower bound the variance $\text{Var}_p(f)$ in terms of $\|f\|_{mon}$.

**Lemma 4.4.** *Fix any $\theta \in \Theta_B$ and define $p := p_\theta$. Define $R := 2^{d+8}nBM$. Then for any $f \in \mathbb{R}[x_1, \ldots, x_n]_{\leq d}$ with $f(0) = 0$, it holds that*

$$\text{Var}_p(f) \geq \frac{1}{2^{2d}(d+1)^{2d}(16e)^{d+1}M^{2d+3}R^{2d^2+2d}(n+B)^{2d}} \|f\|_{mon}^2.$$

See Appendix C for the proof. We are now ready to finish the proof of Theorem 4.1.

***Proof of Theorem 4.1.*** Fix $\theta \in \Theta_B$. Pick any $w \in \mathbb{R}^M$ and define $f(x) = \langle w, T(x) \rangle$. By definition of $\mathcal{I}(\theta)$, we have $\text{Var}_{p_\theta}(f) = w^\top \mathcal{I}(\theta) w$. Moreover, $\|f\|_{mon}^2 = \|w\|_2^2$. Thus, Lemma 4.4 gives us that $w^\top \mathcal{I}(\theta) w \geq (nB)^{-O(d^3)} \|w\|_2^2$, using that $R = 2^{d+8}nBM$ and $M = \binom{n+d}{d}$. The bound $\lambda_{\min}(\mathcal{I}(\theta)) \geq (nB)^{-O(d^3)}$ follows. $\qquad \square$

## 5 Statistical Efficiency of Score Matching

In this section we prove Theorem 1.3. The main technical ingredient is a bound on the restricted Poincaré constants of distributions in $\mathcal{P}_{n,d,B}$. For any fixed $\theta \in \Theta_B$, we show that $C_P$ can be bounded in terms of the *condition number* of the Fisher information matrix $\mathcal{I}(\theta)$. We describe the building blocks of the proof below.

Fix $\theta, w \in \mathbb{R}^{M-1}$ and define $f(x) := \langle w, T(x) \rangle$. First, we need to upper bound $\text{Var}_{p_\theta}(f)$. This is where (the first half of) the condition number appears. Using the crucial fact that the restricted Poincaré constant only considers functions $f$ that are linear in the sufficient statistics, and the definition of $\mathcal{I}(\theta)$, we get the following bound on $\text{Var}_{p_\theta}(f)$ in terms of the coefficient vector $w$. The proof is deferred to Section D.

**Lemma 5.1.** *Fix $\theta, w \in \mathbb{R}^{M-1}$ and define $f(x) := \langle w, T(x) \rangle$. Then*

$$\|w\|_2^2 \, \lambda_{\min}(\mathcal{I}(\theta)) \leq \text{Var}_{p_\theta}(f) \leq \|w\|_2^2 \, \lambda_{\max}(\mathcal{I}(\theta)).$$

Next, we lower bound $\mathbb{E}_{x \sim p_\theta} \|\nabla_x f(x)\|_2^2$. To do so, we could pick an orthonormal basis and bound $\mathbb{E}\langle u, \nabla_x f(x) \rangle^2$ over all directions $u$ in the basis; however, it is unclear how to choose this basis. Instead, we pick $u \sim \mathcal{N}(0, I_n)$ randomly, and use the following identity:

$$\mathbb{E}_{x \sim p_\theta}[\|\nabla_x f(x)\|_2^2] = \mathbb{E}_{x \sim p_\theta} \mathbb{E}_{u \sim N(0, I_n)} \langle u, \nabla_x f(x) \rangle^2$$

For any fixed $u$, the function $g(x) = \langle u, \nabla_x f(x) \rangle$ is also a polynomial. If this polynomial had no constant coefficient, we could immediately lower bound $\mathbb{E}\langle u, \nabla_x f(x) \rangle^2$ in terms of the remaining coefficients, as above. Of course, it may have a nonzero constant coefficient, but with some case-work over the value of the constant, we can still prove the following bound:

**Lemma 5.2.** *Fix $\theta, \tilde{w} \in \mathbb{R}^{M-1}$ and $c \in \mathbb{R}$, and define $g(x) := \langle \tilde{w}, T(x) \rangle + c$. Then*

$$\mathbb{E}_{x \sim p_\theta}[g(x)^2] \geq \frac{c^2 + \|\tilde{w}\|_2^2}{4 + 4 \|\mathbb{E}[T(x)]\|_2^2} \min(1, \lambda_{min}(\mathcal{I}(\theta))).$$

*Proof.* We have

$$\begin{aligned} \mathbb{E}_{x \sim p_\theta}[g(x)^2] &= \text{Var}_{p_\theta}(g) + \mathbb{E}_{x \sim p_\theta}[g(x)]^2 \\ &= \text{Var}_{p_\theta}(g - c) + (c + \tilde{w}^\top \mathbb{E}_{x \sim p_\theta}[T(x)])^2 \\ &\geq \|\tilde{w}\|_2^2 \, \lambda_{\min}(\mathcal{I}(\theta)) + (c + \tilde{w}^\top \mathbb{E}_{x \sim p_\theta}[T(x)])^2 \end{aligned}$$

where the inequality is by Lemma 5.1. We now distinguish two cases.

**Case I.** Suppose that $|c + \tilde{w}^\top \mathbb{E}_{x \sim p_\theta}[T(x)]| \geq c/2$. Then

$$\mathbb{E}_{x \sim p_\theta}[g(x)^2] \geq \|\tilde{w}\|_2^2 \, \lambda_{\min}(\mathcal{I}(\theta)) + \frac{c^2}{4} \geq \frac{c^2 + \|\tilde{w}\|_2^2}{4} \min(1, \lambda_{\min}(\mathcal{I}(\theta))).$$

**Case II.** Otherwise, we have $|c + \tilde{w}^\top \mathbb{E}_{x \sim p_\theta}[T(x)]| < c/2$. By the triangle inequality, it follows that $|\tilde{w}^\top \mathbb{E}_{x \sim p_\theta}[T(x)]| \geq c/2$, so $\|\tilde{w}\|_2 \geq c/(2 \|\mathbb{E}_{x \sim p_\theta}[T(x)]\|_2)$. Therefore

$$c^2 + \|\tilde{w}\|_2^2 \leq (1 + 4 \|\mathbb{E}_{x \sim p_\theta}[T(x)]\|_2^2) \|\tilde{w}\|_2^2,$$

from which we get that

$$\mathbb{E}_{x \sim p_\theta}[g(x)^2] \geq \|\tilde{w}\|_2^2 \, \lambda_{\min}(\mathcal{I}(\theta)) \geq \frac{c^2 + \|\tilde{w}\|_2^2}{1 + 4 \|\mathbb{E}_{x \sim p_\theta}[T(x)]\|_2^2} \lambda_{\min}(\mathcal{I}(\theta))$$

as claimed. $\qquad\square$

With Lemma 5.1 and Lemma 5.2 in hand (taking $g(x) = \langle u, \nabla_x f(x) \rangle$ in the latter), all that remains is to relate the squared monomial norm of $\langle u, \nabla_x f(x) \rangle$ (in expectation over $u$) to the squared monomial norm of $f$. This crucially uses the choice $u \sim N(0, I_n)$. We put together the pieces in the following lemma. The detailed proof is provided in Section D.

**Lemma 5.3.** *Fix $\theta, w \in \mathbb{R}^{M-1}$. Define $f(x) := \langle w, T(x) \rangle$. Then*

$$\text{Var}_{p_\theta}(f) \leq (4 + 4 \|\mathbb{E}_{x \sim p_\theta}[T(x)]\|_2^2) \frac{\lambda_{max}(\mathcal{I}(\theta))}{\min(1, \lambda_{min}(\mathcal{I}(\theta)))} \mathbb{E}_{x \sim p_\theta}[\|\nabla_x f(x)\|_2^2].$$

Finally, putting together Lemma 5.3, Theorem 4.1 (that lower bounds $\lambda_{\min}(\mathcal{I}(\theta))$), and Lemma B.2 (that upper bounds $\lambda_{\max}(\mathcal{I}(\theta))$ – a straightforward consequence of the distributions in $\mathcal{P}_{n,d,B}$ having bounded moments), we can prove the following formal version of Theorem 1.3:

**Theorem 5.4.** *Fix $n, d, B, N \in \mathbb{N}$. Pick any $\theta^* \in \Theta_B$ and let $x^{(1)}, \dots, x^{(N)} \sim p_{\theta^*}$ be independent samples. Then as $N \to \infty$, the score matching estimator $\hat{\theta}_{\mathrm{SM}} = \hat{\theta}_{\mathrm{SM}}(x^{(1)}, \dots, x^{(N)})$ satisfies*

$$\sqrt{N}(\hat{\theta}_{\mathrm{SM}} - \theta^*) \to N(0, \Gamma)$$

*where $\|\Gamma\|_{op} \leq (nB)^{O(d^3)}$. As a corollary, for all sufficiently large $N$ it holds with probability at least $0.99$ that $\left\|\hat{\theta}_{\mathrm{SM}} - \theta^*\right\|_2^2 \leq (nB)^{O(d^3)}/N$.*

*Proof.* We apply Theorem 2.2. By Lemma B.4 and the fact that $\lambda_{\min}(I(\theta^*)) > 0$ (Theorem 4.1), the necessary regularity conditions are satisfied so that the score matching estimator is consistent and asymptotically normal, with asymptotic covariance $\Gamma$ satisfying

$$\|\Gamma\|_{op} \leq \frac{2C_P^2(\|\theta\|_2^2 \, \mathbb{E}_{x \sim p_{\theta^*}} \|(JT)(x)\|_{op}^4 + \mathbb{E}_{x \sim p_{\theta^*}} \|\Delta T(x)\|_2^2)}{\lambda_{\min}(\mathcal{I}(\theta^*))^2} \tag{4}$$

where $C_P$ is the restricted Poincaré constant for $p_{\theta^*}$ with respect to linear functions in $T(x)$ (see Definition 2.1). By Lemma 5.3, we have

$$C_P \leq (4 + 4 \|\mathbb{E}_{x \sim p_\theta}[T(x)]\|_2^2) \frac{\lambda_{\max}(\mathcal{I}(\theta^*))}{\min(1, \lambda_{\min}(\mathcal{I}(\theta^*))}$$

$$\leq (4 + 4B^{2d}M^{2d+2}2^{2d(d+6)+1}) \frac{B^{2d}M^{2d+1}2^{2d(d+6)+1}}{(nB)^{-O(d^3)}} \quad \leq (nB)^{O(d^3)}$$

using parts (a) and (b) of Lemma B.2; Theorem 4.1; and the fact that $M = \binom{n+d}{d}$. Substituting into (4) and bounding the remaining terms using Lemma B.3 and a second application of Theorem 4.1, we conclude that $\|\Gamma\|_{op} \leq (nB)^{O(d^3)}$ as claimed. The high-probability bound now follows from Markov's inequality; see Remark 4 in Koehler et al. (2022) for details. □

# 6 Conclusion

We have provided a concrete example of an exponential family—namely, exponentials of bounded degree polynomials—where score matching is significantly more computationally efficient than maximum likelihood estimation (through optimization with a zero- or first-order oracle), while still achieving the same sample efficiency up to polynomial factors. While score matching was designed to be more computationally efficient for exponential families, the determination of statistical complexity is more challenging, and we give the first separation between these two methods for a general class of functions.

As we have restricted our attention to the asymptotic behavior of both of the methods, an interesting future direction is to see how the finite sample complexities differ. One could also give a more fine-grained comparison between the polynomial dependencies of score matching and MLE, which we have not attempted to optimize. Finally, it would be interesting to relate our results with similar results and algorithms for learning Ising and higher-order spin glass models in the discrete setting, and give a more unified treatment of psuedo-likelihood or score/ratio matching algorithms in these different settings.

# Acknowledgements

CP is supported by Moses Charikar and Greg Valiant's Simons Investigator Awards. DR is supported in part by a U.S. DoD NDSEG Fellowship. AS is supported in part by NSF grants DMS-2054503 and 2238125. AM is supported by a grant from the ONR, NSF Award 1918656 and a David and Lucile Packard Fellowship. AR is supported in part by NSF awards IIS-2211907, CCF-2238523, and an Amazon Research Award.

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

# A Omitted Proofs from Section 3

**Theorem A.1** (Valiant and Vazirani (1985); Cook (1971)). *Suppose that there is a randomized* $\mathrm{poly}(n)$*-time algorithm for the following problem: given a 3-CNF formula* $\mathcal{C}$ *with* $n$ *variables and at most* $5n$ *clauses, under the promise that* $\mathcal{C}$ *has at most one satisfying assignment, determine whether* $\mathcal{C}$ *is satisfiable. Then,* $\mathsf{NP} = \mathsf{RP}$.

**Lemma A.2.** *In the setting of Definition 3.3, set* $d := 7$ *and* $B := 64m\alpha + 2\beta$. *Then* $p_{\mathcal{C},\alpha,\beta} \in \mathcal{P}_{n,d,B}$.

*Proof.* Since $\alpha H_{\mathcal{C}}(x) + \beta G(x)$ is a polynomial in $x_1, \ldots, x_n$ of degree at most 7, there is some $\theta = \theta(\mathcal{C}, \alpha, \beta) \in \mathbb{R}^{M-1}$ such that $\langle \theta, T(x) \rangle + \alpha H_{\mathcal{C}}(x) + \beta G(x)$ is a constant independent of $x$. Then $h(x) \exp(-\alpha H_{\mathcal{C}}(x) - \beta G(x))$ is proportional to $h(x) \exp(\langle \theta, T(x) \rangle)$, so $p_{\mathcal{C},\alpha,\beta} = p_\theta$. Moreover, for any clause $C_j$, every monomial of $H_{C_j}$ has coefficient at most 64 in absolute value, so every monomial of $H_{\mathcal{C}}$ has coefficient at most $64m$. Similarly, every monomial of $G$ has coefficient at most 2 in absolute value. Thus, $\|\theta\|_\infty \leq 64m\alpha + 2\beta =: B$, so $p_{\mathcal{C},\alpha,\beta} \in \mathcal{P}_{n,d,B}$. $\qquad\square$

Given a point $v \in \mathcal{H}$, let $\mathcal{O}(v) := \{x \in \mathbb{R}^n : x_i v_i \geq 0; \forall i \in [n]\}$ denote the octant containing $v$, and let $\mathcal{B}_r(v) := \{x \in \mathbb{R}^n : \|x - v\|_\infty \leq r\}$ denote the ball of radius $r$ with respect to $\ell_\infty$ norm.

**Lemma A.3.** *Let* $p := p_{\mathcal{C},\alpha,\beta}$ *and* $Z := Z_{\mathcal{C},\alpha,\beta}$ *for some 3-CNF* $\mathcal{C}$ *with* $m$ *clauses and* $n$ *variables, and some parameters* $\alpha, \beta > 0$. *Let* $r \in (0, 1)$. *If* $\beta \geq 40r^{-2} \log(4n/r)$, *then for any* $v \in \mathcal{H}$ *that is a satisfying assignment for* $\mathcal{C}$,

$$\Pr_{x \sim p} (x \in \mathcal{B}_r(v)) \geq \frac{e^{-1-81m\alpha r^2}}{Z} \left( \int_0^\infty \exp(-x^8 - \beta(1 - x^2)^2) \, dx \right)^n.$$

*For any* $w \in \mathcal{H}$ *that is not a satisfying assignment for* $\mathcal{C}$,

$$\Pr_{x \sim p} (x \in \mathcal{O}(w)) \leq \frac{e^{-\alpha}}{Z} \left( \int_0^\infty \exp(-x^8 - \beta(1 - x^2)^2) \, dx \right)^n.$$

*Proof.* We begin by lower bounding the probability over $\mathcal{B}_r(v)$. Pick any clause $C_\ell$ included in $\mathcal{C}$. We claim that $H_{C_\ell}(v') \leq 81r^2$ for all $v' \in \mathcal{B}_r(v)$. Indeed, say that $C_\ell = \tilde{x}_i \vee \tilde{x}_j \vee \tilde{x}_k$. Since $v$ satisfies $C_\ell$, at least one of $\{f_i(v_i), f_j(v_j), f_k(v_k)\}$ must be zero. Without loss of generality, say that $f_i(v_i) = 0$; also observe that $|f_j(v_j)|, |f_k(v_k)| \leq 2$. It follows that for any $v' \in \mathcal{B}_r(v)$, $|f_i(v'_i)| \leq r$ and $|f_j(v'_j)|, |f_j(v'_k)| \leq 2 + r \leq 3$ (since $r \leq 1$). Therefore, we have

$$H_{C_\ell}(v') \leq r^2 \cdot (3)^2 \cdot (3)^2 = 81r^2.$$

Summing over all $m$ possible clauses, we have $H_{\mathcal{C}}(v') \leq 81mr^2$ for all $v' \in \mathcal{B}_r(v)$. Hence,

$$
\begin{aligned}
\Pr_{x \sim p} (x \in \mathcal{B}_r(v)) &= \frac{1}{Z} \int_{\mathcal{B}_r(v)} \exp\left( -\sum_{i=1}^n x_i^8 - \alpha H_{\mathcal{C}}(x) - \beta G(x) \right) dx \\
&\geq \frac{e^{-81m\alpha r^2}}{Z} \int_{\mathcal{B}_r(v)} \exp\left( -\sum_{i=1}^n x_i^8 - \beta G(x) \right) dx \\
&= \frac{e^{-81m\alpha r^2}}{Z} \left( \int_{1-r}^{1+r} \exp(-x^8 - \beta(1 - x^2)^2) \, dx \right)^n \\
&\geq \frac{e^{-81m\alpha r^2}}{Z} \left( 1 + \frac{1}{n} \right)^{-n} \left( \int_0^\infty \exp(-x^8 - \beta(1 - x^2)^2) \, dx \right)^n \qquad (5) \\
&\geq \frac{e^{-1-81m\alpha r^2}}{Z} \left( \int_0^\infty \exp(-x^8 - \beta(1 - x^2)^2) \, dx \right)^n
\end{aligned}
$$

where the second inequality (5) is by Lemma A.8. Next, we upper bound the probability over $\mathcal{O}(w)$. Let $C_\ell$ be any clause in $\mathcal{C}$ that is not satisfied by $w$. Say that $C_\ell = \tilde{x}_i \vee \tilde{x}_j \vee \tilde{x}_k$. Then $|f_i(w_i)| = |f_j(w_j)| = |f_k(w_k)| = 2$. Furthermore, for any $w' \in \mathcal{O}^d(w)$, we have

$|f_i(w_i')|, |f_j(w_j')|, |f_k(w_k')| \geq 1$, and hence $H_{C_\ell}(w') \geq 1$. Since $H_{C'}(x) \geq 0$ for all $x, C'$, we conclude that $H_C(w') \geq H_{C_\ell}(w') \geq 1$ for all $w' \in \mathcal{O}(w)$. In particular, this gives us

$$
\begin{aligned}
\Pr_{x \sim p}(x \in \mathcal{O}(w)) &= \frac{1}{Z} \int_{\mathcal{O}(w)} \exp\left(-\sum_{i=1}^n x_i^8 - \alpha H_C(x) - \beta G(x)\right) dx \\
&\leq \frac{e^{-\alpha}}{Z} \int_{\mathcal{O}(w)} \exp\left(-\sum_{i=1}^n x_i^8 - \beta G(x)\right) dx \\
&= \frac{e^{-\alpha}}{Z} \left(\int_0^\infty \exp(-x^8 - \beta(1-x^2)^2)\, dx\right)^n
\end{aligned}
$$

as claimed. $\qquad\square$

### A.1 Hardness of approximate sampling

**Lemma A.4.** *Let $C$ be a satisfiable instance of 3-SAT with $m$ clauses and $n$ variables. Let $\alpha, \beta > 0$ satisfy $\alpha \geq 2(n+1)$ and $\beta \geq 6480m \log(13n\sqrt{m})$. Set $p := p_{C,\alpha,\beta}$ and $Z := Z_{C,\alpha,\beta}$. If $\mathcal{V} \subseteq \mathcal{H}$ is the set of satisfiable assignments for $C$, then*

$$
\sum_{v \in \mathcal{V}} \Pr_{x \sim p}(x \in \mathcal{O}(v)) \geq \frac{1}{2}.
$$

*Proof.* Let $v \in \mathcal{H}$ be any assignment that satisfies $C$, and let $w \in \mathcal{H}$ be any assignment that does not satisfy $C$. By Lemma A.3 with $r = 1/\sqrt{162m}$, we have

$$
\begin{aligned}
\Pr_{x \sim p_C}(x \in \mathcal{O}(v)) &\geq \Pr_{x \sim p_C}(x \in B_r(v)) \\
&\geq \frac{e^{-1-\alpha/2}}{Z} \left(\int_0^\infty \exp(-x^8 - \beta(1-x^2)^2)\, dx\right)^n \\
&\geq e^{-1+\alpha/2} \Pr(x \in \mathcal{O}(w)).
\end{aligned}
$$

Since we chose $\alpha$ sufficiently large that $e^{-1+\alpha/2} \geq 2^n$, we get that

$$
\Pr_{x \sim p_C}(x \in \mathcal{O}(v)) \geq \sum_{w \in \mathcal{H} \setminus \mathcal{V}} \Pr_{x \sim p_C}(x \in \mathcal{O}(w)).
$$

Hence,

$$
\sum_{v \in \mathcal{V}} \Pr_{x \sim p_C}(x \in \mathcal{O}(v)) \geq \sum_{w \in \mathcal{H} \setminus \mathcal{V}} \Pr_{x \sim p_C}(x \in \mathcal{O}(w)) = 1 - \sum_{v \in \mathcal{V}} \Pr_{x \sim p_C}(x \in \mathcal{O}(v)).
$$

The lemma statement follows. $\qquad\square$

**Theorem A.5.** *Let $B \geq Cn^2$ for a sufficiently large constant $C$. Unless RP = NP, there is no algorithm which takes as input an arbitrary $\theta \in \Theta_B$ and outputs a sample from a distribution $Q$ with $\mathrm{TV}(P_\theta, Q) \leq 1/3$ in $\mathrm{poly}(n)$ time.*

*Proof.* Suppose that such an algorithm exists. For each $n \in \mathbb{N}$ define $\alpha = 2(n+1)$ and $\beta = 32400n \log(13n\sqrt{5n})$. Given a 3-CNF formula $C$ with $n$ variables and at most $5n$ clauses, we can compute $\theta = \theta(C, \alpha, \beta)$. By Lemma A.2 we have $\theta \in \Theta_B$ so long as $B \geq Cn^2$ for a sufficiently large constant $C$. Thus, by assumption we can generate a a sample from a distribution $Q$ with $\mathrm{TV}(P_{C,\alpha,\beta}, Q) \leq 1/3$. But by Lemma A.4, we have $\Pr_{x \sim P_{C,\alpha,\beta}}[\mathrm{sign}(x) \text{ satisfies } C] \geq 1/2$. Thus, $\Pr_{x \sim Q}[\mathrm{sign}(x) \text{ satisfies } C] \geq 1/6$. It follows that we can find a satisfying assignment with $O(1)$ invocations of the sampling algorithm in expectation. By Theorem A.1 we get NP = RP. $\qquad\square$

## A.2 Hardness of approximating zeroth-order oracle

**Lemma A.6.** *Fix $n, m \in \mathbb{N}$ and let $\alpha \geq 2(n+1)$ and $\beta \geq 6480m \log(13n\sqrt{m})$. There is a constant $A = A(n, m, \alpha, \beta)$ so that the following hold for every 3-CNF formula $\mathcal{C}$ with $n$ variables and $m$ clauses:*

- *If $\mathcal{C}$ is unsatisfiable, then $Z_{\mathcal{C},\alpha,\beta} \leq A$*
- *If $\mathcal{C}$ is satisfiable, then $Z_{\mathcal{C},\alpha,\beta} \geq (e/2)^n A$.*

*Proof.* If $\mathcal{C}$ is unsatisfiable, then by the second part of Lemma A.3, we have

$$Z = Z \sum_{w \in \mathcal{H}} \Pr_{x \sim p}(x \in \mathcal{O}(w)) \leq 2^n e^{-\alpha} \left( \int_0^\infty \exp(-x^{d+1} - \beta(1-x^2)^2) \, dx \right)^n =: A_{\text{unsat}}.$$

On the other hand, if $\mathcal{C}$ is satisfiable, then by the first part of Lemma A.3 with $r = 1/\sqrt{162m}$,

$$Z \geq Z \Pr_{x \sim p}(x \in \mathcal{B}_r(v)) \geq e^{-1-\alpha/2} \left( \int_0^\infty \exp(-x^{d+1} - \beta(1-x^2)^2) \, dx \right)^n =: A_{\text{sat}}.$$

Since $\alpha \geq 2(n+1)$, we get

$$A_{\text{unsat}} \leq (2/e)^n A_{\text{sat}}$$

as claimed. $\qquad\square$

*Proof of Theorem 3.4.* First, observe that the following problem is NP-hard (under randomized reductions): given two 3-CNF formulas $\mathcal{C}, \mathcal{C}'$ each with $n$ variables and at most $10n$ clauses, where it is promised that exactly one of the formulas is satisfiable, determine which of the formulas is satisfiable. Indeed, this follows from Theorem A.1: given a 3-CNF formula $\mathcal{C}$ with $n$ variables, at most $5n$ clauses, and at most one satisfying assignment, consider adjoining either the clause $x_i$ or the clause $\neg x_i$ to $\mathcal{C}$. If $\mathcal{C}$ has a satisfying assignment $v^*$, then exactly one of the resulting formulas is satisfiable, and determining which one is satisfiable identifies $v_i^*$. Repeating this procedure for all $i \in [n]$ yields an assignment $v$, which satisfies $\mathcal{C}$ if and only if $\mathcal{C}$ is satisfiable.

For each $n \in \mathbb{N}$ define $\alpha = 2(n+1)$ and $\beta = 64800n \log(13n\sqrt{10n})$. Let $B > 0$ be chosen later. Suppose that there is a $\text{poly}(n)$-time algorithm which, given $\theta \in \Theta_B$, computes an approximation of $\log Z_\theta$ with additive error less than $n \log 1.16$. Then given two formulas $\mathcal{C}$ and $\mathcal{C}'$ with $n$ variables and at most $10n$ clauses each, we can compute $\theta = \theta(\mathcal{C}, \alpha, \beta)$ and $\theta' = \theta(\mathcal{C}', \alpha, \beta)$. By Lemma A.2, we have $\theta, \theta' \in \Theta_B$ so long as $B \geq Cn^2$ for a sufficiently large constant $C$. Hence by assumption we can compute approximations $\tilde{Z}_\theta$ and $\tilde{Z}_{\theta'}$ of $Z_\theta$ and $Z_{\theta'}$ respectively, with multiplicative error less than $1.16^n$. However, by Lemma A.6 and the assumption that exactly one of $\mathcal{C}$ and $\mathcal{C}'$ is satisfiable, we know that $\tilde{Z}_\theta > \tilde{Z}_{\theta'}$ if and only if $\mathcal{C}$ is satisfiable. Thus, $\mathsf{NP} = \mathsf{RP}$. $\qquad\square$

## A.3 Hardness of approximating first-order oracle

**Lemma A.7.** *Let $\mathcal{C}$ be a 3-CNF formula with $m$ clauses and $n$ variables, and exactly one satisfying assignment $v^* \in \mathcal{H}$. Let $\alpha = 4n$ and $\beta \geq 25920mn \log(102n\sqrt{mn})$, and define $p := p_{\mathcal{C},\alpha,\beta}$ and $Z := Z_{\mathcal{C},\alpha,\beta}$. Then $\mathbb{E}_{x \sim p}[v_i^* x_i] \geq 1/20$ for all $i \in [n]$.*

*Proof.* Without loss of generality take $i = 1$ and $v_i^* = 1$. Set $r = 1/(\sqrt{648mn})$, $\alpha = 4n$, and $\beta \geq 40r^{-2} \log(4n/r)$. We want to show that $\mathbb{E}_{x \sim p}[x_1] \geq 1/20$. We can write

$$\mathbb{E}[x_1] = \mathbb{E}[x_1 \mathbb{1}[x \in B_r(v^*)]] + \mathbb{E}[x_1 \mathbb{1}[x \in \mathcal{O}(v^*) \setminus B_r(v^*)]] + \sum_{v \in \mathcal{H} \setminus \{v^*\}} \mathbb{E}[x_1 \mathbb{1}[x \in \mathcal{O}(v)]]$$

$$\geq (1-r) \Pr[x \in B_r(v^*)] - 2^n \max_{v \in \mathcal{H} \setminus \{v^*\}} \mathbb{E}[|x_1| \mathbb{1}[x \in \mathcal{O}(v)]] \tag{6}$$

since $x_1 \geq 1 - r$ for $x \in B_r(v^*)$ and $x_1 \geq 0$ for $x \in \mathcal{O}(v^*)$. Now observe that on the one hand,

$$\Pr(x \in B_r(v^*)) \geq \frac{e^{-1-81m\alpha r^2}}{Z} \left( \int_0^\infty \exp(-x^* - \beta g(x)) \, dx \right)^n \tag{7}$$

by Lemma A.3. On the other hand, for any $v \in \mathcal{H} \setminus \{v^*\}$,

$$
\begin{aligned}
\mathbb{E}[|x_1| \mathbb{1}[x \in \mathcal{O}(v)]] &= \frac{1}{Z} \int_{\mathcal{O}(v)} |x_1| \exp\left( -\sum_{i=1}^{n} x_i^8 - \alpha H(x) - \beta G(x) \right) dx \\
&\leq \frac{e^{-\alpha}}{Z} \int_{\mathcal{O}(v)} |x_1| \exp\left( -\sum_{i=1}^{n} x_i^8 - \beta G(x) \right) dx \\
&= \frac{e^{-\alpha}}{Z} \left( \int_0^\infty x \exp(-x^8 - \beta g(x)) \, dx \right) \left( \int_0^\infty \exp(-x^8 - \beta g(x)) \, dx \right)^{n-1} \\
&\leq \frac{2e^{-\alpha}}{Z} \left( \int_0^\infty \exp(-x^8 - \beta g(x)) \, dx \right)^n \quad (8)
\end{aligned}
$$

where the second inequality is by Lemma A.9 with $k = 1$. Combining (7) and (8) with (6), we have

$$
\begin{aligned}
\mathbb{E}[x_1] &\geq \frac{(1-r)e^{-1-81m\alpha r^2} - 2^{n+1}e^{-\alpha}}{Z} \left( \int_0^\infty \exp(-x^8 - \beta g(x)) \, dx \right)^n \\
&\geq \frac{1}{10Z} \left( \int_0^\infty \exp(-x^8 - \beta g(x)) \, dx \right)^n \\
&\geq \frac{1}{10Z} \int_{\mathcal{O}(v^*)} \exp\left( -\sum_{i=1}^{n} x_i^8 - \alpha H(x) - \beta G(x) \right) dx \\
&= \frac{1}{10} \Pr[x \in \mathcal{O}(v^*)] \\
&\geq \frac{1}{20}
\end{aligned}
$$

where the second inequality is by choice of $\alpha$ and $r$; the third inequality is by nonnegativity of $H(x)$; and the fourth inequality is by Lemma A.4 and uniqueness of the satisfying assignment $v^*$. $\qquad\square$

*Proof of Theorem 3.5.* Suppose that such an algorithm exists. Set $\alpha = 4n$ and $\beta = 129600n^2 \log(102n^2\sqrt{5})$. Given a 3-CNF formula $\mathcal{C}$ with $n$ variables, at most $5n$ clauses, and exactly one satisfying assignment $v^* \in \mathcal{H}$, we can compute $\theta = \theta(\mathcal{C}, \alpha, \beta)$. Let $E \in \mathbb{R}^n$ be the algorithm's estimate of $\nabla_\theta \log Z_\theta = \mathbb{E}_{x \sim p_{\mathcal{C},\alpha,\beta}} T(x)$. Then $\left\| E - \mathbb{E}_{x \sim p_{\mathcal{C},\alpha,\beta}} T(x) \right\|_\infty < 1/20$. But by Lemma A.7, for each $i \in [n]$, the $i$-th entry of $\mathbb{E}_{x \sim p_{\mathcal{C},\alpha,\beta}} T(x)$, which corresponds to the monomial $x_i$, has sign $v_i^*$ and magnitude at least $1/20$. Thus, $\text{sign}(E_i) = v_i^*$. So we can compute $v^*$ in polynomial time. By Theorem A.1, it follows that $\mathsf{NP} = \mathsf{RP}$. $\qquad\square$

## A.4 Integral bounds

**Lemma A.8.** *Fix $\beta > 150$ and $\gamma \in [0, 1]$. Define $f : \mathbb{R} \to \mathbb{R}$ by $f(x) = \gamma x^8 + \beta(1 - x^2)^2$. Pick any $r \in (6/\beta, 0.04)$. Then*

$$
\int_0^\infty \exp(-f(x)) \, dx \leq \left( \frac{1}{1 - \exp(-\beta r^2/8)} + \frac{2\exp(-\beta r/40)}{r} \right) \int_{1-r}^{1+r} \exp(-f(x)) \, dx.
$$

*In particular, for any $m \in \mathbb{N}$, if $\beta \geq 40r^{-2} \log(4m/r)$, then*

$$
\int_0^\infty \exp(-f(x)) \, dx \leq \left( 1 + \frac{1}{m} \right) \int_{1-r}^{1+r} \exp(-f(x)) \, dx.
$$

*Proof.* Set $a = 1/\sqrt{2}$. For any $x \in [a, \infty)$ we have $f''(x) = 56\gamma x^6 - 4\beta + 12\beta x^2 \geq \beta > 0$ for $\beta > 150$. Thus, $f$ has at most one critical point in $[a, \infty)$; call this point $t_0$. Since $f'(x) = 8\gamma x^7 - 4\beta x(1-x^2)$, we have $f'(1) = 8\gamma \geq 0$ and $f'(1-3/\beta) \leq 8 - 4\beta(1-3/\beta)(3/\beta)(2-3/\beta) < 0$. Thus, $t_0 \in (1 - 3/\beta, 1]$. Set $r' = r - 3/\beta \geq r/2$. Then

$$
\int_{1-r}^{1+r} \exp(-f(x)) \, dx \geq \int_{t_0-r'}^{t_0+r'} \exp(-f(x)) \, dx.
$$

For every $t \in \mathbb{R}$ define $I(t) = \int_t^{t+r'} \exp(-f(x)) \, dx$. Since $f$ is $\beta$-strongly convex on $[a, \infty)$, we have for any $t \geq t_0$ that

$$f(t + r') - f(t) \geq r' f'(t) + \frac{r'^2}{2} \beta \geq \frac{r'^2}{2} \beta$$

where the final inequality is because $f'(t) \geq 0$ for $t \in [t_0, \infty)$. Thus, for any $t \geq t_0$,

$$I(t + r') = \int_{t+r'}^{t+2r'} \exp(-f(x)) \, dx = \int_t^{t+r} \exp(-f(x + r')) \, dx \leq \exp(-\beta r'^2/2) I(t).$$

By induction, for any $k \in \mathbb{N}$ it holds that $I(t_0 + kr') \leq \exp(-\beta k r'^2/2) I(t_0)$, so

$$\int_{t_0}^{\infty} \exp(-f(x)) \, dx = \sum_{k=0}^{\infty} I(t_0 + kr') \leq I(t_0) \sum_{k=0}^{\infty} \exp(-\beta k r'^2/2) = \frac{I(t_0)}{1 - \exp(-\beta r'^2/2)}. \quad (9)$$

Similarly, for any $t \in [a + r', t_0]$, we have

$$f(t - r') - f(t) \geq -r' f'(t) + \frac{r'^2}{2} \beta \geq \frac{r'^2}{2} \beta$$

using $\beta$-strong convexity on $[a, \infty)$ and the bound $f'(t) \leq 0$ on $[a, t_0]$. Thus, for any $t \in [a, t_0 - r']$,

$$I(t - r') = \int_{t-r'}^t \exp(-f(x)) \, dx = \int_t^{t+r'} \exp(-f(x - r')) \, dx \leq \exp(-\beta r'^2/2) I(t),$$

so by induction, $I(t_0 - kr') \leq \exp(-\beta(k-1)r'^2/2) I(t_0 - r')$ for any $1 \leq k \leq K := \lfloor (t_0 - a)/r' \rfloor$. It follows that

$$\int_{t_0 - Kr'}^{t_0} \exp(-f(x)) \, dx = \sum_{k=1}^K I(t_0 - kr') \leq I(t_0 - r') \sum_{k=1}^K \exp(-\beta(k-1)r'^2/2) \leq \frac{I(t_0 - r')}{1 - \exp(-\beta r'^2/2)}.$$
$$(10)$$

Finally, note that $t_0 - (K - 1)r' \leq a + 2r' \leq 0.8$. For any $x \in [0, 0.8]$, we have $f'(x) \leq 8x^7 - 0.72\beta x = x(8x^6 - 1.44\beta) \leq 0$, since $\beta > 150$. That is, $f$ is non-increasing on $[0, t_0 - (K - 1)r']$. It follows that

$$\int_0^{t_0 - Kr'} \exp(-f(x)) \, dx \leq \frac{t_0 - Kr'}{r'} \int_{t_0 - Kr'}^{t_0 - (K-1)r'} \exp(-f(x)) \, dx$$
$$\leq \frac{1}{r'} I(t_0 - Kr')$$
$$\leq \frac{\exp(-\beta(K-1)r'^2/2)}{r'} I(t_0 - r').$$

Since $(K - 1)r' \geq t_0 - 0.8 \geq 1 - \frac{3}{\beta} - 0.8 \geq 0.1$, we conclude that

$$\int_0^{t_0 - Kr'} \exp(-f(x)) \, dx \leq \frac{\exp(-\beta r'/20)}{r'} I(t_0 - r'). \quad (11)$$

Combining (9), (10), and (11), we get

$$\int_0^{\infty} \exp(-f(x)) \, dx \leq \left( \frac{1}{1 - \exp(-\beta r'^2/2)} + \frac{\exp(-\beta r'/20)}{r'} \right) \int_{t_0 - r'}^{t_0 + r'} \exp(-f(x)) \, dx.$$

Substituting in $r' \geq r/2$ gives the claimed result. $\qquad \square$

**Lemma A.9.** *Fix $\beta \geq 160 \log(8)$. Then for any $1 \leq k \leq 8$,*

$$\int_0^{\infty} x^k \exp(-x^8 - \beta(1 - x^2)^2) \, dx \leq 2^k \int_0^{\infty} \exp(-x^8 - \beta(1 - x^2)^2) \, dx.$$

*Proof.* Define a distribution $q(x) \propto \exp(-x^8 - \beta(1-x^2)^2)$ for $x \in [0, \infty)$. We want to show that $\mathbb{E}_q[x^k] \leq 2^k$. Indeed,

$$
\begin{aligned}
\mathbb{E}_q[\exp(x^8)] &= \frac{\int_0^\infty \exp(-\beta(1-x^2)^2)\, dx}{\int_0^\infty \exp(-x^8 - \beta(1-x^2)^2)\, dx} \\
&\leq \frac{2 \int_{1/2}^{3/2} \exp(-\beta(1-x^2)^2)\, dx}{\int_0^\infty \exp(-x^8 - \beta(1-x^2)^2)\, dx} \\
&= 2\mathbb{E}_q[\exp(x^8)\mathbb{1}[1/2 \leq x \leq 3/2]] \\
&\leq 2\exp((3/2)^8)
\end{aligned}
$$

where the first inequality is by an application of Lemma A.8 with $r = 1/2$ and $m = 1$. Now by Jensen's inequality we get

$$
\mathbb{E}_q[x^8] \leq \log \mathbb{E}_q[\exp(x^8)] = \log(2) + (3/2)^8 \leq 2^8
$$

and consequently, an application of Hölder inequality gives us $\mathbb{E}_q[x^k] \leq 2^k$, for any $1 \leq k \leq 8$. $\square$

## A.5 Proof of Corollary 3.6

***Proof of Corollary 3.6.*** Recall that $\log p_\theta(x) = \log h(x) + \langle \theta, T(x) \rangle - \log Z_\theta$. Therefore $L_{\text{MLE}}(\theta) = \hat{\mathbb{E}} \log h(x) + \langle \theta, \hat{\mathbb{E}}T(x) \rangle - \log Z_\theta$ and $\nabla_\theta L_{\text{MLE}}(\theta) = \hat{\mathbb{E}}T(x) - \nabla_\theta \log Z_\theta$. Note that we can compute $\hat{\mathbb{E}} \log h(x)$ and $\hat{\mathbb{E}}T(x)$ exactly. It follows that if we can approximate $L_{\text{MLE}}(\theta)$ up to an additive error of $n \log 1.16$, then we can compute $\log Z_\theta$ up to an additive error of $n \log 1.16$. Similarly, if we can compute $\nabla_\theta L_{\text{MLE}}(\theta)$ up to an additive error of $1/20$, then we can compute $\nabla_\theta \log Z_\theta$ up to an additive error of $1/20$. This contradicts Theorems 3.4 and 3.5 respectively, completing the proof. $\square$

# B Moment bounds

**Lemma B.1** (Moment bound). *For any $\theta \in \Theta_B$, $i \in [n]$, and $\ell \in \mathbb{N}$ it holds that*

$$
\mathbb{E}_{x \sim p_\theta} x_i^\ell \leq 32^\ell \max(2\ell^\ell, B^\ell M^\ell 2^{\ell(d+1)+1}).
$$

*Proof.* Without loss of generality assume $i = 1$. Let $L_0 := 32 \max(\ell, BM2^{d+1})$. Then

$$
\begin{aligned}
\mathbb{E}_{x \sim p_\theta} x_1^\ell &\leq L_0^\ell + \mathbb{E}_{x \sim p_\theta} x_1^\ell \mathbb{1}[\|x\|_\infty > L_0] \\
&= L_0^\ell + \sum_{k=0}^\infty \mathbb{E}_{x \sim p_\theta} \left[ x_1^\ell \mathbb{1}[2^k L_0 < \|x\|_\infty \leq 2^{k+1} L_0] \right]
\end{aligned}
$$

Now for any $L \geq L_0$,

$$
\begin{aligned}
&\mathbb{E}\left[ x_1^\ell \mathbb{1}[L < \|x\|_\infty \leq 2L] \right] \\
&= \frac{1}{Z_\theta} \int_{B_{2L}(0) \setminus B_L(0)} x_1^\ell \exp\left( -\sum_{i=1}^n x_i^{d+1} + \langle \theta, T(x) \rangle \right) dx \\
&\leq \frac{(2L)^n}{Z_\theta} (2L)^\ell \exp\left( -L^{d+1} + BM(2L)^d \right) \\
&\leq \frac{(2L)^{n+\ell} \exp(-L^{d+1}/2)}{Z_\theta}.
\end{aligned}
$$

We can lower bound $Z_\theta$ as

$$
\begin{aligned}
Z_\theta &\geq \int_{\mathcal{B}_{1/(BM)}(0)} \exp\left( -\sum_{i=1}^n x_i^{d+1} + \langle \theta, T(x) \rangle \right) dx \\
&\geq (BM)^{-n} \exp(-n(BM)^{-d-1} - BM(BM)^{-1}) \\
&\geq e^{-2}(BM)^{-n}.
\end{aligned}
$$

Thus,

$$\mathbb{E}\left[x_1^\ell \mathbb{1}[L < \|x\|_\infty \le 2L]\right] \le \exp\left((n+\ell)\log(2L) - \frac{1}{2}L^{d+1} + 2 + n\log(BM)\right)$$

$$\le \exp\left(-\frac{1}{4}L^{d+1}\right)$$

where the last inequality uses the facts that $L \ge L_0$, $B \ge 1$, and $M \ge n$ to get

$$(n+\ell)\log(2L) + 2 + n\log(BM) \le 2\max(n,\ell) \cdot L + \frac{L^2}{16} + \frac{L^2}{16} \le \frac{3L^2}{16} \le \frac{L^{d+1}}{4}.$$

We conclude that

$$\mathbb{E}_{x\sim p_\theta}x_1^\ell \le L_0^\ell + \sum_{k=0}^\infty \exp\left(-\frac{1}{4}2^{k(d+1)}L_0^{d+1}\right)$$

$$\le L_0^\ell + 1 \qquad \le 2L_0^\ell$$

which completes the proof. $\qquad\square$

**Lemma B.2** (Largest eigenvalue bound). *For any $\theta \in \Theta_B$, it holds that*

$$\mathbb{E}_{x\sim p_\theta}T(x)T(x)^\top \preceq B^{2d}M^{2d+1}2^{2d(d+6)+1}.$$

*We also have the following consequences:*

(a) $\|\mathbb{E}_{x\sim p_\theta}T(x)\|_2^2 \le B^{2d}M^{2d+2}2^{2d(d+6)+1}$,

(b) $\lambda_{max}(\mathcal{I}(\theta)) \le B^{2d}M^{2d+1}2^{2d(d+6)+1}$,

(c) $\Pr_{x\sim p_\theta}[\|x\|_\infty > 2^{d+8}nBM] \le 1/2$.

*Proof.* Fix any $u, v \in [M]$. Then $T(x)_u T(x)_v = \prod_{i=1}^n x_i^{\gamma_i}$ for some nonnegative integers $\gamma_1, \ldots, \gamma_n$ where $d' := \sum_{i=1}^n \gamma_i \le 2d$. Therefore

$$\mathbb{E}_{x\sim p_\theta}T(x)_u T(x)_v = \mathbb{E}_{x\sim p_\theta}\prod_{i=1}^n x_i^{\gamma_i} \le \prod_{i=1}^n \left(\mathbb{E}_{x\sim p_\theta}x_i^{d'}\right)^{\gamma_i/d'} \le 32^{2d}B^{2d}M^{2d}2^{2d(d+1)+1}$$

by Holder's inequality and Lemma B.1 (with $\ell = 2d$). The claimed spectral bound follows. To prove (a), observe that

$$\|\mathbb{E}_{x\sim p_\theta}T(x)\|_2^2 \le \mathbb{E}_{x\sim p_\theta}\|T(x)\|_2^2 = \text{Tr}\,\mathbb{E}_{x\sim p_\theta}T(x)T(x)^\top \le M\lambda_{max}(\mathbb{E}_{x\sim p_\theta}T(x)T(x)^\top)$$

To prove (b), observe that $\mathcal{I}(\theta) \preceq \mathbb{E}_{x\sim p_\theta}T(x)T(x)^\top$. To prove (c), observe that for any $i \in [n]$,

$$\Pr_{x\sim p_\theta}[|x_i| > 2^{d+8}nBM] \le \frac{\mathbb{E}_{x\sim p_\theta}x_i^{2d}}{(2^{d+8}nBM)^{2d}} \le \frac{1}{2n}.$$

A union bound over $i \in [n]$ completes the proof. $\qquad\square$

**Lemma B.3** (Smoothness bounds). *For every $\theta \in \Theta_B$, it holds that*

$$\mathbb{E}_{x\sim p_\theta}\|\Delta T(x)\|_2^2 := \sum_{j=1}^M \mathbb{E}_{x\sim p_\theta}(\Delta T_j(x))^2 \le n^2 d^4 B^{4d}M^{4d+3}2^{4d(d+6)+2}$$

*and*

$$\mathbb{E}_{x\sim p_\theta}\|(JT)(x)\|_{op}^2 \le n^2 d^4 B^{4d}M^{4d+2}2^{4d(d+6)+2}.$$

*Proof.* Fix any $j \in [M]$; then there is a degree function $\mathbf{d}$ with $1 \le |\mathbf{d}| \le d$ so that $T_j(x) = x_{\mathbf{d}} = \prod_{i=1}^n x_i^{\mathbf{d}(i)}$. Therefore

$$\Delta T_j(x) = \sum_{k\in[n]:\mathbf{d}(k)\ge 2} \mathbf{d}(k)(\mathbf{d}(k) - 1)x_{\mathbf{d}-2\{k\}} =: \langle w, T(x)\rangle$$

for some $w \in \mathbb{R}^M$ with $\|w\|_2^2 = \sum_{k \in [n]:\mathbf{d}(k) \geq 2} \mathbf{d}(k)^2 (\mathbf{d}(k) - 1)^2 \leq d^4$. By Corollary B.2, we conclude that

$$\mathbb{E}_{x \sim p_\theta}(\Delta T_j(x))^2 = \mathbb{E}_{x \sim p_\theta} \langle w, T(x) \rangle^2 \leq n^2 d^4 B^{4d} M^{4d+2} 2^{4d(d+6)+1}.$$

Summing over $j \in [M]$ gives the first claimed bound. For the second bound, observe that

$$\mathbb{E}_{x \sim p_\theta} \|(JT)(x)\|_{\mathsf{op}}^4 \leq \mathbb{E}_{x \sim p_\theta} \|(JT)(x)\|_F^4 = \mathbb{E}_{x \sim p_\theta} \left( \sum_{j=1}^M \sum_{i=1}^n \left( \frac{\partial}{\partial x_i} T_j(x) \right)^2 \right)^2.$$

For any $j \in [M]$ and $i \in [n]$, there is some degree function $\mathbf{d}$ with $|\mathbf{d}| \leq d$ and $\frac{\partial}{\partial x_i} T_j(x) = |\mathbf{d}| \cdot x_{\mathbf{d} - \{i\}}$. Thus, by Holder's inequality and Lemma B.1 (with $\ell = 4d$), we get

$$\mathbb{E}_{x \sim p_\theta} \left( \sum_{j=1}^M \sum_{i=1}^n \left( \frac{\partial}{\partial x_i} T_j(x) \right)^2 \right)^2 = \sum_{j,j' \in [M]} \sum_{i,i' \in [n]} \mathbb{E}_{x \sim p_\theta} \left( \frac{\partial}{\partial x_i} T_j(x) \right)^2 \left( \frac{\partial}{\partial x_{i'}} T_{j'}(x) \right)^2$$

$$\leq M^2 n^2 d^4 B^{4d} M^{4d} 2^{4d(d+6)+2}$$

which proves the second bound. $\square$

The following regularity conditions are sufficient for consistency and asymptotic normality of score matching, assuming that the restricted Poincaré constant is finite and $\lambda_{\min}(\mathcal{I}(\theta^*)) > 0$ (see Proposition 2 in Forbes and Lauritzen (2015) together with Lemma 1 in Koehler et al. (2022)). We show that these conditions hold for our chosen exponential family.

**Lemma B.4** (Regularity conditions). *For any $\theta \in \mathbb{R}^M$, the quantities $\mathbb{E}_{x \sim p_\theta} \|\nabla h(x)\|_2^4$, $\mathbb{E}_{x \sim p_\theta} \|\Delta T(x)\|_2^2$, and $\mathbb{E}_{x \sim p_\theta} \|(JT)(x)\|_{\mathsf{op}}^4$ are all finite. Moreover, $p_\theta(x) \to 0$ and $\|\nabla_x p_\theta(x)\|_2 \to 0$ as $\|x\|_2 \to \infty$.*

*Proof.* Both of the quantities $\|\nabla h(x)\|_2^4$ and $\|\Delta T(x)\|_2^2$ can be written as a polynomial in $x$. Finiteness of the expectation under $p_\theta$ follows from Holder's inequality and Lemma B.1 (with parameter $B$ set to $\|\theta\|_\infty$). Finiteness of $\mathbb{E}_{x \sim p_\theta} \|(JT)(x)\|_{\mathsf{op}}^4$ is shown in Lemma B.3 (again, with $B := \|\theta\|_\infty$). The decay condition $p_\theta(x) \to 0$ holds because $\log p_\theta(x) + \log Z_\theta = -\sum_{i=1}^n x_i^{d+1} + \langle \theta, T(x) \rangle$. For $x \in \mathbb{R}^n$ with $L \leq \|x\|_\infty \leq 2L$, the RHS is at most $-L^{d+1} + M \|\theta\|_\infty (2L)^d$, which goes to $-\infty$ as $L \to \infty$. A similar bound shows that $\|\nabla_x p_\theta(x)\|_2 \to 0$. $\square$

## C  Omitted Proofs from Section 4

***Proof of Lemma 4.2.*** We use the fact that the Legendre polynomials

$$L_k(x) = \frac{1}{2^k} \sum_{j=0}^k \binom{k}{j}^2 (x-1)^{k-j}(x+1)^j,$$

for integers $0 \leq k \leq d$, form an orthogonal basis for the vector space $\mathbb{R}[x]_{\leq d}$ with respect to $L^2[-1,1]$ (see e.g. Koepf (1998)). We consider the normalized versions $\hat{L}_k = \sqrt{\frac{2k+1}{2}} L_k$, so that $\left\| \hat{L}_k \right\|_{L^2[-1,1]} = 1$. By tensorization, the set of products of Legendre polynomials

$$\hat{L}_\mathbf{d}(x) = \prod_{i=1}^n \hat{L}_{\mathbf{d}(i)}(x_i),$$

as $\mathbf{d}$ ranges over degree functions with $|\mathbf{d}| \leq d$, form an orthonormal basis for $\mathbb{R}[x_1, \ldots, x_n]_{\leq d}$ with respect to $L^2([-1,1]^n)$.

Using the formula for $L_k$, we obtain that the sum of absolute values of coefficients of $L_k$ (in the monomial basis) is at most $\frac{1}{2^k} \sum_{j=0}^k \binom{k}{j}^2 2^k \leq 2^{2k}$. By the bound $\|\cdot\|_2 \leq \|\cdot\|_1$ and the definition of $\hat{L}_k$,

$$\left\| \hat{L}_k \right\|_{\mathsf{mon}}^2 \leq \frac{2k+1}{2} \|L_k\|_{\mathsf{mon}}^2 \leq \frac{2k+1}{2} 2^{4k}$$

and hence for any degree function $\mathbf{d}$ with $|\mathbf{d}| \leq d$,

$$\left\|\hat{L}_{\mathbf{d}}\right\|_{\text{mon}}^2 = \prod_{i=1}^n \left\|\hat{L}_{\mathbf{d}(i)}\right\|_{\text{mon}}^2 \leq \prod_{i=1}^n \frac{2\mathbf{d}(i) + 1}{2} 2^{4\mathbf{d}(i)}$$

$$\leq \prod_{i=1}^n e^{\mathbf{d}(i)} 2^{4\mathbf{d}(i)} \leq (16e)^d.$$

Consider any polynomial $f \in \mathbb{R}[x_1, \ldots, x_n]_{\leq d}$, and write $f = \sum_{|\mathbf{d}| \leq d} a_{\mathbf{d}} \hat{L}_{\mathbf{d}}$. By orthonormality, it holds that $\sum_{|\mathbf{d}| \leq d} a_{\mathbf{d}}^2 = \|f\|_{L^2([-1,1]^n)}^2$. Thus, by the triangle inequality and Cauchy-Schwarz,

$$\|p\|_{\text{mon}}^2 = \left\|\sum_{|\mathbf{d}| \leq d} a_{\mathbf{d}} \hat{L}_{\mathbf{d}}\right\|_{\text{mon}}^2 \leq \sum_{|\mathbf{d}| \leq d} a_{\mathbf{d}}^2 \cdot \sum_{|\mathbf{d}| \leq d} \left\|\hat{L}_{\mathbf{d}}\right\|_{\text{mon}}^2$$

$$\leq \|p\|_{L^2([-1,1]^n)}^2 \binom{n + d}{d} (16e)^d$$

as claimed. $\qquad\qquad\square$

***Proof of Lemma 4.3.*** Let $f \in \mathbb{R}[x_1, \ldots, x_n]_{\leq d}$ be a polynomial of degree at most $d$ in $x_1, \ldots, x_n$. Define $g(x) = f(x) - \mathbb{E}_{x \sim p} f(x)$. Set $\epsilon = 1/(2(d + 1)MR^d(n + B))$ and let $(W_i)_{i \in I}$ be $\ell_\infty$-balls of radius $\epsilon$ partitioning $\{x \in \mathbb{R}^n : \|x\|_\infty \leq R\}$. Define random variable $X \sim p|\{\|X\|_\infty \leq R\}$ and let $\iota \in I$ be the random index so that $X \in W_\iota$. Then

$$\text{Var}_p(f) = \mathbb{E}_{x \sim p}[g(x)^2]$$

$$\geq \frac{1}{2}\mathbb{E}[g(X)^2]$$

$$= \frac{1}{2}\mathbb{E}_\iota \mathbb{E}_X[g(X)^2 | X \in W_\iota]$$

where the inequality uses guarantee (c) of Lemma B.2 that $\Pr_{x \sim p}[\|x\|_\infty > R] \leq 1/2$.

Thus, there exists some $\iota^* \in I$ such that $\mathbb{E}_X[g(X)^2 | X \in W_{\iota^*}] \leq 2\text{Var}_p(f)$. Let $q : \mathbb{R}^n \to \mathbb{R}_+$ be the density function of $X | X \in W_{\iota^*}$. Since $q(x) \propto p(x)\mathbb{1}[x \in W_{\iota^*}]$, for any $u, v \in W_{\iota^*}$ we have that

$$\frac{q(u)}{q(v)} = \frac{p(u)}{p(v)} = \frac{h(u)\exp(\langle\theta, T(u)\rangle)}{h(v)\exp(\langle\theta, T(v)\rangle)}$$

$$= \exp\left(\sum_{i=1}^n v_i^{d+1} - u_i^{d+1} + \langle\theta, T(u) - T(v)\rangle\right).$$

Applying Lemma C.1, we get that

$$\frac{q(u)}{q(v)} \leq \exp\left(n(d + 1)R^d \|u - v\|_\infty + MB \|T(u) - T(v)\|_\infty\right)$$

$$\leq \exp\left((n + B) \cdot M(d + 1)R^d \|u - v\|_\infty\right)$$

$$\leq \exp(2\epsilon(n + B) \cdot M(d + 1)R^d)$$

$$\leq \exp(1)$$

by choice of $\epsilon$. It follows that if $\tilde{\mathcal{U}}$ is the uniform distribution on $W_{\iota^*}$, then $q(x) \geq e^{-1}\tilde{\mathcal{U}}(x)$ for all $x \in \mathbb{R}^n$. Thus,

$$\text{Var}_p(f) \geq \frac{1}{2}\mathbb{E}_X[g(X)^2 | X \in W_{\iota^*}] \geq \frac{1}{2e}\mathbb{E}_{x \sim \tilde{\mathcal{U}}}[g(x)^2] \geq \frac{1}{2e}\text{Var}_{\tilde{\mathcal{U}}}(g) = \frac{1}{2e}\text{Var}_{\tilde{\mathcal{U}}}(f)$$

as desired. $\qquad\qquad\square$

**Lemma C.1.** *Fix $R > 0$. For any degree function $\mathbf{d} : [n] \to \mathbb{N}$ with $|\mathbf{d}| \leq d$, and for any $u, v \in \mathbb{R}^n$ with $\|u\|_\infty, \|v\|_\infty \leq R$, it holds that*

$$|u_{\mathbf{d}} - v_{\mathbf{d}}| \leq dR^{d-1} \|u - v\|_\infty.$$

*Proof.* Define $m(x) = x_{\mathbf{d}} = \prod_{i=1}^{n} x_i^{\mathbf{d}(i)}$. Then

$$|m(u) - m(v)| \leq \|u - v\|_\infty \sup_{x \in \mathcal{B}_R(0)} \|\nabla_x m(x)\|_1$$

$$= \|u - v\|_\infty \sup_{x \in \mathcal{B}_R(0)} \sum_{i \in [n]:\mathbf{d}(i)>0} \mathbf{d}(i) \prod_{j=1}^{n} x_i^{\mathbf{d}(i)-\mathbb{1}[i=j]}$$

$$\leq \|u - v\|_\infty \cdot dR^{d-1}$$

as claimed. $\square$

***Proof of Lemma 4.4.*** By Lemma 4.3, there is some $z \in \mathbb{R}^n$ with $\|z\|_\infty \leq R$ and some $\epsilon \geq 1/(2(d+1)MR^d(n+B))$ so that if $\tilde{\mathcal{U}}$ is the uniform distribution on $\{x \in \mathbb{R}^n : \|x - z\|_\infty \leq \epsilon\}$, then

$$\text{Var}_p(f) \geq \frac{1}{2e} \text{Var}_{\tilde{\mathcal{U}}}(f).$$

Define $g : \mathbb{R}^n \to \mathbb{R}$ by $g(x) = f(\epsilon x + z) - \mathbb{E}_{\tilde{\mathcal{U}}} f$. Then by Lemma 4.2,

$$\|g\|_{\text{mon}}^2 \leq (16e)^d M \mathbb{E}_{x \sim \text{Unif}([-1,1]^n)} g(x)^2.$$
$$= (16e)^d M \text{Var}_{\tilde{\mathcal{U}}}(f)$$
$$\leq (16e)^{d+1} M \text{Var}_p(f).$$

Write $f(x) = \sum_{1 \leq |\mathbf{d}| \leq d} \alpha_{\mathbf{d}} x_{\mathbf{d}}$ and $g(x) = \sum_{1 \leq |\mathbf{d}| \leq d} \beta_{\mathbf{d}} x_{\mathbf{d}}$. We know that $f(x) = g(\epsilon^{-1}(x - z)) + \mathbb{E}_{\tilde{\mathcal{U}}} f$. Thus, for any nonzero degree function $\mathbf{d}$, we have

$$\alpha_{\mathbf{d}} = \sum_{\substack{\mathbf{d}' \geq \mathbf{d} \\ |\mathbf{d}'| \leq d}} \epsilon^{-|\mathbf{d}'|}(-z)^{\mathbf{d}'-\mathbf{d}} \beta_{\mathbf{d}'}.$$

Thus $|\alpha_{\mathbf{d}}| \leq \epsilon^{-d} R^d \|\beta\|_1 \leq \epsilon^{-d} R^d \sqrt{M} \|g\|_{\text{mon}}$, and so summing over monomials gives

$$\|f\|_{\text{mon}}^2 \leq M^2 \epsilon^{-2d} R^{2d} \|g\|_{\text{mon}}^2 \leq (16e)^{d+1} M^3 \epsilon^{-2d} R^{2d} \text{Var}_p(f).$$

Substituting in the choice of $\epsilon$ from Lemma 4.3 completes the proof. $\square$

# D  Omitted Proofs from Section 5

***Proof of Lemma 5.1.*** We have

$$\text{Var}_{p_\theta}(f) = \mathbb{E}_{x \sim p_\theta}[f(x)^2] - \mathbb{E}_{x \sim p_\theta}[f(x)]^2$$
$$= w^\top \mathbb{E}_{x \sim p_\theta}[T(x)T(x)^\top] w - w^\top \mathbb{E}_{x \sim p_\theta}[T(x)] \mathbb{E}_{x \sim p_\theta}[T(x)]^\top w$$
$$= w^\top \mathcal{I}(\theta) w,$$

and since

$$\|w\|_2^2 \lambda_{\min}(\mathcal{I}(\theta)) \leq w^\top \mathcal{I}(\theta) w \leq \|w\|_2^2 \lambda_{\max}(\mathcal{I}(\theta),$$

the lemma statement follows. $\square$

***Proof of Lemma 5.3.*** Since $f(x) = \sum_{1 \leq |\mathbf{d}| \leq d} w_{\mathbf{d}} x_{\mathbf{d}}$, we have for any $u \in \mathbb{R}^n$ that

$$\langle u, \nabla_x f(x) \rangle = \sum_{i=1}^{n} u_i \sum_{0 \leq |\mathbf{d}| < d} (1 + \mathbf{d}(i)) w_{\mathbf{d}+\{i\}} x_{\mathbf{d}} = c(u) + \sum_{1 \leq |\mathbf{d}| < d} \tilde{w}(u)_{\mathbf{d}} x_{\mathbf{d}}$$

where $c(u) := \sum_{i=1}^{n} u_i w_{\{i\}}$ and $\tilde{w}(u)_{\mathbf{d}} := \sum_{i=1}^{n} u_i(1 + \mathbf{d}(i)) w_{\mathbf{d}+\{i\}}$. But now

$$\mathbb{E}_{x \sim p_\theta}[\|\nabla_x f(x)\|_2^2] = \mathbb{E}_{x \sim p_\theta} \mathbb{E}_{u \sim N(0,I_n)} \langle u, \nabla_x f(x) \rangle^2$$
$$= \mathbb{E}_{u \sim N(0,I_n)} \mathbb{E}_{x \sim p_\theta} (c(u) + \langle \tilde{w}(u), T(x) \rangle)^2$$
$$\geq \mathbb{E}_{u \sim N(0,I_n)} \frac{c(u)^2 + \|\tilde{w}(u)\|_2^2}{4 + 4 \|\mathbb{E}_{x \sim p_\theta}[T(x)]\|_2^2} \min(1, \lambda_{\min}(\mathcal{I}(\theta))).$$

where the last inequality is by Lemma 5.2. Finally,

$$
\mathbb{E}_{u \sim N(0, I_n)} \left[ c(u)^2 + \|\tilde{w}(u)\|_2^2 \right] = \sum_{0 \leq |\mathbf{d}| < d} \mathbb{E}_{u \sim N(0, I_n)} \left[ \left( \sum_{i=1}^{n} u_i (1 + \mathbf{d}(i)) w_{\mathbf{d}+\{i\}} \right)^2 \right]
$$

$$
= \sum_{0 \leq |\mathbf{d}| < d} \sum_{i=1}^{n} (1 + \mathbf{d}(i))^2 w_{\mathbf{d}+\{i\}}^2 \qquad \geq \|w\|_2^2
$$

where the second equality is because $\mathbb{E}[u_i u_j] = \mathbb{1}[i = j]$ for all $i, j \in [n]$, and the last inequality is because every term $w_{\mathbf{d}}^2$ in $\|w\|_2^2$ appears in at least one of the terms of the previous summation (and has coefficient at least one). Putting everything together gives

$$
\mathbb{E}_{x \sim p_\theta}[\|\nabla_x f(x)\|_2^2] \geq \frac{\|w\|_2^2}{4 + 4\|\mathbb{E}_{x \sim p_\theta}[T(x)]\|_2^2} \min(1, \lambda_{\min}(\mathcal{I}(\theta)))
$$

$$
\geq \frac{1}{4 + 4\|\mathbb{E}[T(x)]\|_2^2} \frac{\min(1, \lambda_{\min}(\mathcal{I}(\theta)))}{\lambda_{\max}(\mathcal{I}(\theta))} \mathrm{Var}_{p_\theta}(f)
$$

where the last inequality is by Lemma 5.1. $\qquad \square$

