# Provable benefits of score matching

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

\|_{\mathsf{mon}}^2 \leq (4e)^d M \mathbb{E}_{x \sim \mathrm{Unif}([-1,1]^n)} g(x)^2.$$
$$= (4e)^d M \mathrm{Var}_{\tilde{\mathcal{U}}}(f)$$
$$\leq (4e)^{d+1} M \mathrm{Var}_p(f).$$

Write $f(x) = \sum_{1 \leq |\mathbf{d}| \leq d} \alpha_\mathbf{d} x_\mathbf{d}$ and $g(x) = \sum_{1 \leq |\mathbf{d}| \leq d} \beta_\mathbf{d} x_\mathbf{d}$. We know that $f(x) = g(\epsilon^{-1}(x-z)) + \mathbb{E}_{\tilde{\mathcal{U}}}f$. Thus, for any nonzero degree function $\mathbf{d}$, we have

$$\alpha_\mathbf{d} = \sum_{\substack{\mathbf{d}' \geq \mathbf{d} \\ |\mathbf{d}'| \leq d}} \epsilon^{-|\mathbf{d}'|}(-z)^{\mathbf{d}'-\mathbf{d}}\beta_{\mathbf{d}'}.$$

Thus $|\alpha_\mathbf{d}| \leq \epsilon^{-d}R^d\|\beta\|_1 \leq \epsilon^{-d}R^d\sqrt{M}\|g\|_{\mathsf{mon}}$, and so summing over monomials gives

$$\|f\|_{\mathsf{mon}}^2 \leq M^2\epsilon^{-2d}R^{2d}\|g\|_{\mathsf{mon}}^2 \leq (4e)^{d+1}M^3\epsilon^{-2d}R^{2d}\mathrm{Var}_p(f).$$

Substituting in the choice of $\epsilon$ from Lemma 4.3 completes the proof. $\qquad\square$

# D   Omitted Proofs from Section 5

**Proof of Lemma 5.1.** We have

$$\mathrm{Var}_{p_\theta}(f) = \mathbb{E}_{x \sim p_\theta}[f(x)^2] - \mathbb{E}_{x \sim p_\theta}[f(x)]^2$$
$$= w^\top \mathbb{E}_{x \sim p_\theta}[T(x)T(x)^\top]w - w^\top \mathbb{E}_{x \sim p_\theta}[T(x)]\mathbb{E}_{x \sim p_\theta}[T(x)]^\top w$$
$$= w^\top \mathcal{I}(\theta)w,$$

and since

$$\|w\|_2^2 \lambda_{\min}(\mathcal{I}(\theta)) \leq w^\top \mathcal{I}(\theta)w \leq \|w\|_2^2 \lambda_{\max}(\mathcal{I}(\theta)),$$

the lemma statement follows. $\qquad\square$

**Proof of Lemma 5.3.** Since $f(x) = \sum_{1 \leq |\mathbf{d}| \leq d} w_\mathbf{d} x_\mathbf{d}$, we have for any $u \in \mathbb{R}^n$ that

$$\langle u, \nabla_x f(x)\rangle = \sum_{i=1}^n u_i \sum_{0 \leq |\mathbf{d}| < d} (1 + \mathbf{d}(i))w_{\mathbf{d}+\{i\}}x_\mathbf{d} = c(u) + \sum_{1 \leq |\mathbf{d}| < d} \tilde{w}(u)_\mathbf{d} x_\mathbf{d}$$

where $c(u) := \sum_{i=1}^n u_i w_{\{i\}}$ and $\tilde{w}(u)_\mathbf{d} := \sum_{i=1}^n u_i(1 + \mathbf{d}(i))w_{\mathbf{d}+\{i\}}$. But now

$$\mathbb{E}_{x \sim p_\theta}[\|\nabla_x f(x)\|_2^2] = \mathbb{E}_{x \sim p_\theta}\mathbb{E}_{u \sim N(0,I_n)}\langle u, \nabla_x f(x)\rangle^2$$
$$= \mathbb{E}_{u \sim N(0,I_n)}\mathbb{E}_{x \sim p_\theta}(c(u) + \langle \tilde{w}(u), T(x)\rangle)^2$$
$$\geq \mathbb{E}_{u \sim N(0,I_n)}\frac{c(u)^2 + \|\tilde{w}(u)\|_2^2}{4 + 4\|\mathbb{E}_{x \sim p_\theta}[T(x)]\|_2^2}\min(1, \lambda_{\min}(\mathcal{I}(\theta))).$$

where the last inequality is by Lemma 5.2. Finally,

$$\mathbb{E}_{u \sim N(0,I_n)}\left[c(u)^2 + \|\tilde{w}(u)\|_2^2\right] = \sum_{0 \leq |\mathbf{d}| < d}\mathbb{E}_{u \sim N(0,I_n)}\left[\left(\sum_{i=1}^n u_i(1 + \mathbf{d}(i))w_{\mathbf{d}+\{i\}}\right)^2\right]$$
$$= \sum_{0 \leq |\mathbf{d}| < d}\sum_{i=1}^n (1 + \mathbf{d}(i))^2 w_{\mathbf{d}+\{i\}}^2 \qquad \geq \|w\|_2^2$$

where the second equality is because $\mathbb{E}[u_i u_j] = \mathbb{1}[i = j]$ for all $i, j \in [n]$, and the last inequality is because every term $w_{\mathbf{d}}^2$ in $\|w\|_2^2$ appears in at least one of the terms of the previous summation (and has coefficient at least one). Putting everything together gives

$$\mathbb{E}_{x \sim p_\theta}[\|\nabla_x f(x)\|_2^2] \geq \frac{\|w\|_2^2}{4 + 4\|\mathbb{E}_{x \sim p_\theta}[T(x)]\|_2^2} \min(1, \lambda_{\min}(\mathcal{I}(\theta)))$$

$$\geq \frac{1}{4 + 4\|\mathbb{E}[T(x)]\|_2^2} \frac{\min(1, \lambda_{\min}(\mathcal{I}(\theta)))}{\lambda_{\max}(\mathcal{I}(\theta))} \mathrm{Var}_{p_\theta}(f)$$

where the last inequality is by Lemma 5.1. $\qquad\square$