# OpenReview forum: "Provable benefits of score matching"
_NeurIPS.cc/2023/Conference — NeurIPS 2023 spotlight_

### Official Review · Reviewer_BCHk · 2023-06-20

**Soundness:** 4 excellent
**Presentation:** 4 excellent
**Contribution:** 3 good
**Rating:** 7
**Confidence:** 4

**Summary:**

The authors describe an exponential family where the sufficient statistic $T(x)$ contains all non-constant monomials of degree $\leq d$, the background density is $h(x) = \exp(-\sum_{i=1}^n x_i^{d+1})$, and the parameter $\theta$ is constrained to have infinity norm bounded by $B$. Using a reduction from $3\mathsf{SAT}$, they show that under this exponential family, finding the MLE is NP hard, and thus unless $\mathsf{NP} = \mathsf{RP}$, that it would take time exponential in the random vector dimension $n$ to compute the MLE.

They also show that the MLE has asymptotic sample efficiency $(nB)^{O(d^3)}$. They show that score matching also has asymptotic sample efficiency $(nB)^{O(d^3)}$. Unlike the MLE, score matching can be solved in time polynomial in the dimension of $\theta$, which if $d$ is considered constant, is polynomial in $n$. This is because the objective corresponding to score matching is convex in $\theta$.

This is a concrete example of a situation where score matching has a provable advantage over the MLE (same asymptotic sample efficiency, but one needs exponential time unless $\mathsf{NP} = \mathsf{RP}$ and the other is polynomial time).

**Strengths:**

- clear contribution
- relevant related work is discussed
- this is a significant result (rigorous justification for potential benefit of score matching over MLE)

**Weaknesses:**

No glaring weaknesses, but the paper was a bit difficult to follow; I found it hard to connect the math-heavy theorem/lemma statements together. For example, I did not understand how Lemmas 4.2 and 4.3 built toward Lemma 4.4 on my initial read.

Typos spotted:
- Equation under line 172, $\|\theta\|_2^2$ should be $\|\theta^*\|_2^2$?
- Line 482, $f''(x) = 56\gamma x^6 - 2\beta + 6\beta x^2$ should be $f''(x) = 56\gamma x^6 - 4\beta + 12\beta x^2$ I think, but this doesn't affect anything
- Lemma A.6, the $(2/e)^n$ should be replaced with $(e/2)^n$
- Math at the bottom of page 16, second line, you want to minimize the exponent so instead of $-BM(BM)^{-d}$ it should be $-BM(BM)^{-1}$ right? I don't think this affects anything though since you bound it by $-1$ regardless
- Math under line 519, I'm not sure that $(n+\ell)\log(2L) + 2 + n\log(BM) \leq \frac14 L^{d+1}$, unless some lower bound on $d$ is assumed? Like I'm not sure if that holds for $d=1$?
- Lemma B.3, bound on Laplacian in the Lemma statement doesn't seem to match the derivation (exponent of 2d vs 4d)
- Line 564 - this appears to be claiming that $\sum_{j=0}^k \binom{k}{j}^2 = 2^k$, which is false? I think you could bound it by $2^{2k}$ though by Cauchy-Schwarz, so it probably doesn't matter, just changes some constants down the line.
- Line 573 - $B_\iota \to W_\iota$
- Line 583 - $\alpha_i \to d_i$

**Questions:**

- Why is a nonzero constant not allowed for the polynomial defined by the sufficient statistic of the exponential family? It wasn't clear to me, also why that was a requirement for Lemma 4.4 to hold. Does that cause a problem?

- This does not appear necessary to have in this work to me, but how close are we to obtaining finite sample bounds for score matching, analogous to the ones (I assume there are?) for MLE?

---

> ### Author Rebuttal · Authors · 2023-08-09
>
> We thank the reviewer for their time and for a very careful reading. We'll fix the typos -- as the reviewer notes, a few constants will have to be updated but the main results are all unchanged (up to some constant factors in the exponents). To be a bit more precise about the more mathematical typos:
>
> **Q:** *``Math at the bottom of page 16''* You're entirely right, it should be $-BM(BM)^{-1}$; as you say this doesn't change anything.
>
> **Q:** *``Math under line 519"* Instead of $L_0 := \max(\ell, BM2^{d+1})$ we should define $L_0 := 32\max(\ell,BM2^{d+1})$. Now in line 519, since $L \geq L_0$ and $B \geq 1$ and $M \geq n$, we can show that $\frac{1}{4}L^{d+1} \geq (n+\ell)\log(2L) + 2 + n\log(BM)$. Indeed $(n+\ell)\log(2L) \leq 2\max(n,\ell) \cdot L \leq L^2/16$. Similarly $2 \leq L^2/16$ and $n\log(BM) \leq L^2/16$. Finally $L^2 \leq L^{d+1}$. This adds a constant factor of $32^\ell$ to the moment bound in the lemma statement, which is not substantive.
>
> **Q:** *``Lemma B.3''* Yes, thanks, we'll update the statement to match the derivation (the exponents are off by a factor of $2$).
>
> **Q:** *``Line 564''* Yes, since each binomial is at most $2^k$ we can bound the stated quantity by $2^{2k}$, which changes a constant in the exponent.
>
> **Q:** *``Why is a nonzero constant not allowed for the polynomial defined by the sufficient statistic of the exponential family? It wasn't clear to me, also why that was a requirement for Lemma 4.4 to hold. Does that cause a problem?''* Including a nonzero constant as a sufficient statistic wouldn't actually change the family of distributions captured, since an additive constant in the exponent of the density is canceled out by the constant of proportionality. Note that as a result the parameter corresponding to the constant statistic would not even be statistically identifiable. This shows up in Lemma 4.4, where if we allowed e.g. the constant polynomial $f=1$, this has variance $0$ but monomial norm $1$ so the lemma would not be true.
>
> **Q:** *``This does not appear necessary to have in this work to me, but how close are we to obtaining finite sample bounds for score matching, analogous to the ones (I assume there are?) for MLE?''* We believe that it's probably possible to get analogous finite-sample guarantees by similar techniques. In particular, the prior work by Koehler, Hecket, and Risteski (which shows that the asymptotic efficiency of SM and MLE are related via a restricted Poincare constant) also shows that the finite-sample efficiency can be bounded in terms of a restricted log Sobolev constant and a Rademacher bound. It's certainly not immediate from our results, but we would speculate that our techniques for bounding the restricted Poincare constant can extend to bounding the restricted log Sobolev constant, and that the Rademacher complexity should be bounded via standard arguments. It would then remain to convert the KL-divergence error bound (obtained in Theorem 1 of their work) into a parameter error bound, which ought to follow from our bounds on the Fisher information.

---

> > ### Comment · Reviewer_BCHk · 2023-08-16
> >
> > Thank you for taking the time to address my comments!

---

### Official Review · Reviewer_xxrR · 2023-07-06

**Soundness:** 3 good
**Presentation:** 3 good
**Contribution:** 3 good
**Rating:** 7
**Confidence:** 3

**Summary:**

This paper provides an example of fitting exponential family models for which score matching
and MLE are both statistically efficient, but MLE is computationally hard to optimize.

**Strengths:**

The strength is in the construction of an example to showcase the benefit of score matching
over MLE.

**Weaknesses:**

Several aspects  could be improved.

1. First, for the computational lower bound, the points to evaluation loss and gradient is worst case.
When one has samples from such distribution, is solving MLE still computationally hard?

2. Although the aim of this paper is to provide examples to advocate for score matching, but for the
distribution family proposed in the paper, are there simple (or simpler) estimators that are both computationally and
statistically efficient?

3. In the paper, the analysis of the statistical performance is quite crude, which makes it hard to see what's
the statistical cost one needs to pay when computation is the restriction. Is there proper computation-statistic
tradeoff for the model considered in the paper?

**Questions:**

1. Why constant is removed from the construction of the sufficient statistics?

---

> ### Author Rebuttal · Authors · 2023-08-09
>
> We thank the reviewer for their time and comments. To address their questions:
>
> **Q:** *``First, for the computational lower bound, the points to evaluation loss and gradient is worst case. When one has samples from such distribution, is solving MLE still computationally hard?''* Good question; unfortunately proving NP-hardness of average-case problems (i.e. when the input is drawn from a nice distribution) is generally not doable. One alternative is to prove a reduction from some conjecturally average-case hard problem such as Planted Clique, but average-case reductions are notoriously tricky. A final alternative common in theoretical computer science is to prove failure of some restricted class of algorithms (e.g. low-degree polynomials).
>
> All these alternatives are potentially doable, but would become technically involved and stray from the crux of the matter \--- with enough samples, score matching and MLE are essentially just two different techniques for solving the same computational problem, so our main goal with the worst-case computational lower bounds is to illustrate computational difficulties faced by a standard implementation/analysis of MLE.
>
> **Q:** *``Although the aim of this paper is to provide examples to advocate for score matching, but for the distribution family proposed in the paper, are there simple (or simpler) estimators that are both computationally and statistically efficient?''* We are not aware of any other computationally and statistically efficient estimators for this family. Perhaps the closest related work is the following paper, which suggests a computationally efficient estimator for learning some exponential family distributions. However, they require the family to have bounded support, in addition to several (rather complex) norm assumptions.
>
> Shah, Shah, and Wornell. \emph{``A Computationally Efficient Method for Learning Exponential Family Distributions".}
>
> **Q:** *``In the paper, the analysis of the statistical performance is quite crude, which makes it hard to see what's the statistical cost one needs to pay when computation is the restriction. Is there proper computation-statistic tradeoff for the model considered in the paper?''* Understanding the statistical cost of score matching at a more fine-grained level is an interesting direction for future work. It is possible that score matching matches the statistical efficiency MLE, although it seems more likely there is some polynomial gap. Speaking more broadly, we are not aware of *any* continuous exponential family with a provable statistical/computational tradeoff. Of course, proving such a lower bound is orthogonal to the purpose of this work, which was simply to show that score matching is roughly as efficient as MLE without the computational drawbacks. But this is still useful context and helps explain why proving lower bounds and separations is challenging even for exponential families. We will add this discussion.
>
> **Q:** *``Why constant is removed from the construction of the sufficient statistics?''* Including a nonzero constant as a sufficient statistic wouldn't actually change the family of distributions captured, since an additive constant in the exponent of the density is canceled out by the constant of proportionality. Note that as a result the parameter corresponding to the constant statistic would not even be statistically identifiable. This shows up in Lemma 4.4, where if we allowed e.g. the constant polynomial $f=1$, this has variance $0$ but monomial norm $1$ so the lemma would not be true.

---

> > ### Comment · Reviewer_xxrR · 2023-08-15
> >
> > Thanks for the response to address my questions.

---

### Official Review · Reviewer_bZrt · 2023-07-06

**Soundness:** 4 excellent
**Presentation:** 3 good
**Contribution:** 2 fair
**Rating:** 6
**Confidence:** 3

**Summary:**

In this paper, the authors present a mathematical setting where Score Matching (SM) method has more statistical benefits than Maximum Likelihood (ML) technique, when estimating a parameterized probability distribution $p_\theta \in P(\mathbb{R}^n)$ known up to a normalizing constant $Z_\theta$. In particular, they describe an explicit exponential family of distributions $F$ for which SM loss $L_{SM}$ is efficient to compute, with same statistical efficiency as ML loss $L_{ML}$ (Theorem 2 and 3), while $L_{ML}$ is shown to be intractable in polynomial time depending on the parameters of this family (Theorem 1). Given an odd integer $d$ and $B>0$, any distribution $p_\theta \in F$ is notably defined by (i) its vector of sufficient statistics, which consists of all monomials in $x^1,..., x_n$ of at least degree $1$ and at most degree $d$, and (ii) its parameter $\theta$, which lies in the $\ell_\infty$-ball with radius $B$. This work is the first one to put in perspective the statistical efficiency of SM and ML for a large family of continuous probability distributions.


Three main theoretical results are stated here. In Theorem 1, the authors prove that, for any $p_\theta \in F$ (with $d=7$) and any set of $N$ independent samples from $p_\theta$, it is NP-hard (in $n$ and $N$) to provide an accurate approximation of $L_{ML}(\theta)$ and $\nabla L_{ML}(\theta)$. This result comes from the difficulty to approximate $Z_\theta$, which is necessary to compute $L_{ML}(\theta)$, while it does not appear in $L_{SM}(\theta)$. In Theorem 2, they derive an upper bound of the $\ell_2$-error between $\theta$ and its ML estimator obtained via $N$ samples, in the limit where $N\to \infty$, for any $p_\theta \in F$. Their proof relies on the asymptotic result given by [1] and consists of lower bounding the smallest eigenvalue of the Fisher information matrix of $p_\theta$. In Theorem 3, they derive the same upper bound in the case of the SM estimator for any $p_\theta \in F$, by invoking the asymptotic result from [2] and bounding the Poincaré constant of $p_\theta$. Combined together, Theorems 2 and 3 show that ML and SM techniques roundly have the same statistical efficiency.

[1] Asymptotic statistics, Van der Vaart, 2000.

[2] Statistical Efficiency of Score Matching: The View from Isoperimetry, Koehler et al., 2022.

**Strengths:**

- This work provides a fair comparison between statistical efficiency for SM and ML technique, which is of primary interest for the community of machine learning.
- Although Theorems 2 and 3 rely on important theoretical results, their results are not straightforward to obtain, and their proofs are original and clear to understand.

**Weaknesses:**

Although the theoretical results presented here are interesting in their own, they should be combined with numerical experiments, which should illustrate the trade-off between these two methods in practice, depending on the parameters of the exponential family.

**Questions:**

Why do the authors do not keep the whole dependence in $d$ in the bound for Theorems 2 and 3 ? Does it change something between SM and ML ?

**Limitations:**

In this work, the asymptotic theory is elaborated on the fact that the MLE and SM estimators can be computed exactly from a collection of samples. However, to compare these two techniques for practical purpose, one should include approximation error on the estimator.

---

> ### Author Rebuttal · Authors · 2023-08-09
>
> We thank the reviewer for their time and comments. To address the reviewer's questions:
>
> **Q:** *``Why do the authors do not keep the whole dependence in $d$ in the bound for Theorems 2 and 3 ? Does it change something between SM and ML ?''* In both cases the dependence is $O(d^3)$. We stated the dependence as $\textsf{poly}(d)$ in the introduction just to make the theorems cleaner \--- we are thinking of $d$ as a small constant, and the key takeaway is that for both score matching and maximum likelihood, the sample complexity is $\textsf{poly}(n,B)$. It's certainly plausible that maximum likelihood may achieve a smaller constant in the exponent than score matching; proving such a statistical separation is an interesting direction for future work but complementary to the purpose of this work, which is to establish conditions under which they achieve comparable rates.
>
> **Q:** *``In this work, the asymptotic theory is elaborated on the fact that the MLE and SM estimators can be computed exactly from a collection of samples. However, to compare these two techniques for practical purpose, one should include approximation error on the estimator.''* It's true that there may be some approximation error in computing the MLE from finite samples (even ignoring our evidence that this task may be computationally intractable). However, as shown in Equation (2) in our paper, the score matching estimator has a closed form in our setting, so there is no approximation error.

---

> > ### Comment · Reviewer_bZrt · 2023-08-14
> >
> > Thank you for your response. Although I find the theoretical result really interesting and non-trivial, I think that there should be numerical experiments that support the theory. That is why I keep my score unchanged.

---

### Official Review · Reviewer_fYcf · 2023-07-06

**Soundness:** 4 excellent
**Presentation:** 3 good
**Contribution:** 3 good
**Rating:** 7
**Confidence:** 3

**Summary:**

The paper attempts to elucidate the theoretical reasoning for the benefits seen in score matching. The author proposes a family of exponential distributions that can efficiently compute the score matching loss while having a comparable statistical efficiency to that of maximum likelihood.

**Strengths:**

- The paper is well written/organized and easy to follow. The author does a good job introducing related theoretical information.
- Equations are extensively described with a thorough description.

**Weaknesses:**

- There are no experimental results. It would be nice if there were some machine learning models that were optimized with score matching and ML and compared with each other.

**Questions:**

- What is the benefit of using this score matching method than taking the denoising score matching approach?
- Is this approach more computational efficient than denoising score matching? How well those this score matching approach scale for high dimensional data?

**Limitations:**

The author has adequately addressed limitations.

---

> ### Author Rebuttal · Authors · 2023-08-09
>
> We thank the reviewer for their time and comments. To address the reviewer's questions:
>
> **Q:** *``What is the benefit of using this score matching method than taking the denoising score matching approach?''* Recall that the latter is score matching applied to an annealed version of the distribution. It's generally believed that the annealing improves the statistical performance of score matching. It's also generally believed that score matching (even without annealing) has computational benefits over MLE. The goal of this paper was to provide a provable example of the latter. The fact that annealing may further improve statistical efficiency in a sense only strengthens the thesis of our paper: that the score matching technique has provable computational benefits over the standard MLE technique, without sacrificing statistical efficiency.
>
> **Q:** *``Is this approach more computational efficient than denoising score matching? How well those this score matching approach scale for high dimensional data?"* For exponential families, the score matching loss is quadratic, so the global optimum actually has a closed-form expression (see equation (2) in the paper), which can be computed in time polynomial in the number of sufficient statistics and linear in the number of samples. In high dimensional settings one might be interesting in even stronger computational guarantees, but we have not yet investigated whether this is possible, and if it is would likely require new techniques.
>
> It's possible that the denoising score matching loss could also be optimized efficiently, but it's not immediately clear one way or the other.

---

> > ### Comment · Reviewer_fYcf · 2023-08-21
> >
> > Thank you for your response. I will keep my score the same.

---

### Decision · Program_Chairs · 2023-09-21

**Decision:**

Accept (spotlight)

**Comment:**

This paper theoretically explains the success of score matching by showing that it _can_ be simultaneously computationally and statistically efficient, when the MLE is not. Reviewers agree that this is a significant contribution that is worth sharing with the community. That said, there are some suggestions that could considerably improve the paper. First, a visualization of the results either in terms of synthetic experiments or computational/statistical behavioral comparison between MLE and SM, would help better appreciate the theory. Second, even though prior methods to establish efficient learning make additional assumptions (e.g. Shah et al. 21), a better comparison would be fruitful. Third, the narrative and presentation could be improved. The significance of the main and intermediate results are indeed explained. However, a more top to bottom strategy could help appreciate the analysis: instead of listing lemmas then the theorem, it could help to sketch the theorem first and highlight where the lemmas are needed, and then state them. The authors are also encouraged to incorporate the other minor comments and suggestions raised by the reviewers.